# Evolution of *lasR* mutants in polymorphic *Pseudomonas aeruginosa* populations facilitates chronic infection of the lung

Kelei Zhao [1] ✉, Xiting Yang[1], Qianglin Zeng[1], Yige Zhang[2], Heyue Li[3], Chaochao Yan[4], Jing Shirley Li[2], Huan Liu[2], Liangming Du[1], Yi Wu[1], Gui Huang[1], Ting Huang[1], Yamei Zhang[1], Hui Zhou[1], Xinrong Wang[1], Yiwen Chu [1] ✉ & Xikun Zhou [2] ✉

Chronic infection with the bacterial pathogen *Pseudomonas aeruginosa* often leads to coexistence of heterogeneous populations carrying diverse mutations. In particular, loss-of-function mutations affecting the quorum-sensing regulator LasR are often found in bacteria isolated from patients with lung chronic infection and cystic fibrosis. Here, we study the evolutionary dynamics of polymorphic *P. aeruginosa* populations using isolates longitudinally collected from patients with chronic obstructive pulmonary disease (COPD). We find that isolates deficient in production of different sharable extracellular products are sequentially selected in COPD airways, and *lasR* mutants appear to be selected first due to their quorum-sensing defects. Polymorphic populations including *lasR* mutants display survival advantages in animal models of infection and modulate immune responses. Our study sheds light on the multistage evolution of *P. aeruginosa* populations during their adaptation to host lungs.

Microbes evolve diverse capacities to colonize virtually all ecosystems and reserve the possibilities to transfer into new habitats. As unicellular organism with simple cell structure, the success of bacterial colonization in different environments is largely attributed to the execution of group behaviors, typically social cooperation[1,2]. Bacterial cooperative interactions are mainly mediated by costly and sharable extracellular products (public goods), and can unite the local individuals to increase population fitness in specific habitats[3,4].

*Pseudomonas aeruginosa* is a ubiquitous Gram-negative opportunistic bacterium capable of colonizing a wide range of environments, including the lungs of immunocompromised patients, and a model species in studying the evolution and maintenance of cooperation[3,5–7]. *P. aeruginosa* colonizing host lungs may encounter various environmental pressures, such as low nutrient availability, inflammatory responses, antibiotic treatment, and inter/intraspecific competitions[8–10]. The large genome size and complicated regulatory network of *P. aeruginosa* endow the bacterium with the capacity to engage in a variety of cooperative interactions, such as the extracellular protease-mediated acquisition of macromolecular nutrients, extracellular siderophore-mediated chelation of irons and exopolysaccharide-related biofilm formation[5,6,11,12]. Especially, the quorum-sensing (QS) system, which is mainly composed of three core regulatory cascades with *las* sits atop of *rhl* and *pqs*, positively controls the expression of a large set of extracellular products for the virulence, resistance, and immune evasion of *P. aeruginosa* in the host and the development of social behaviors[7,13,14].

[1]Antibiotics Research and Re-evaluation Key Laboratory of Sichuan Province, School of Pharmacy, Affiliated Hospital of Chengdu University, Chengdu University, 610106 Chengdu, China. [2]Department of Biotherapy, Cancer Center and State Key Laboratory of Biotherapy, West China Hospital, Sichuan University, 610041 Chengdu, China. [3]Key Laboratory of Bio-resources and Eco-environment, Ministry of Education, College of Life Sciences, Sichuan University, 610064 Chengdu, China. [4]Ecological Restoration and Biodiversity Conservation Key Laboratory of Sichuan Province, Chengdu Institute of Biology, Chinese Academy of Sciences, 610041 Chengdu, China. ✉e-mail: zhaokelei@cdu.edu.cn; chuyiwen@cdu.edu.cn; xikunzhou@scu.edu.cn

According to the theory of the classic public goods game, cooperation is vulnerable to the invasion of cheaters who do not produce public goods but benefit from the cooperation of producers (cooperators). Cheating as the optimal strategy will ultimately overtake cooperation and results in the tragedy of the commons due to the shortage of crucial public goods[15,16] (Fig. 1a). However, population collapse is rarely detected in bacterial populations that colonize natural environments or host tissues[3,5,6]. *P. aeruginosa* isolates with loss-of-function mutations in the gene encoding the central regulator of QS system, LasR, are the most frequent in the lungs of patients with the genetic disease cystic fibrosis (CF)[17,18]. Moreover, among the three core regulatory genes of *P. aeruginosa* QS system, only the *lasR* mutant can massively invade the wild-type (WT) strains in a cheating manner under experimental conditions that QS-controlled extracellular proteases are the key limiting factor for cell growth[6,19].

Clinical evidence shows that CF airways are frequently colonized by a mixture of *P. aeruginosa* isolates with intact and mutated *lasR* gene[17,20,21]. Several empirical and mathematical studies have provided conceptual explanations for the maintenance of cooperation in the *P. aeruginosa* population, such as social policing, metabolic prudence, and cheating on cheaters[9,22–24]. Nevertheless, the cooperative interactions of *P. aeruginosa* in vivo might be multifactorial and more complicated. Few studies have explored the evolutionary dynamics of polymorphic *P. aeruginosa* population comprising of *lasR*-intact and *lasR* mutant individuals in host lungs. As *lasR* mutants have a competitive advantage over WT cooperators and some evolved *lasR* mutants can also elicit hyperinflammatory responses[5,25], we hypothesize that *lasR* mutants might be continuously evolved in the polymorphic *P. aeruginosa* population and play important role in facilitating the persistent colonization of the population (Fig. 1a).

In this study, *P. aeruginosa* isolates were longitudinally collected from the bronchoalveolar lavage (BAL) fluids of patients with chronic obstructive pulmonary disease (COPD), followed by phenotypic, genetic, phylogenetic, and multi-omics-based functional analyses (Fig. 1b). We found that the isolates deficient in producing different kinds of sharable extracellular products were sequentially selected

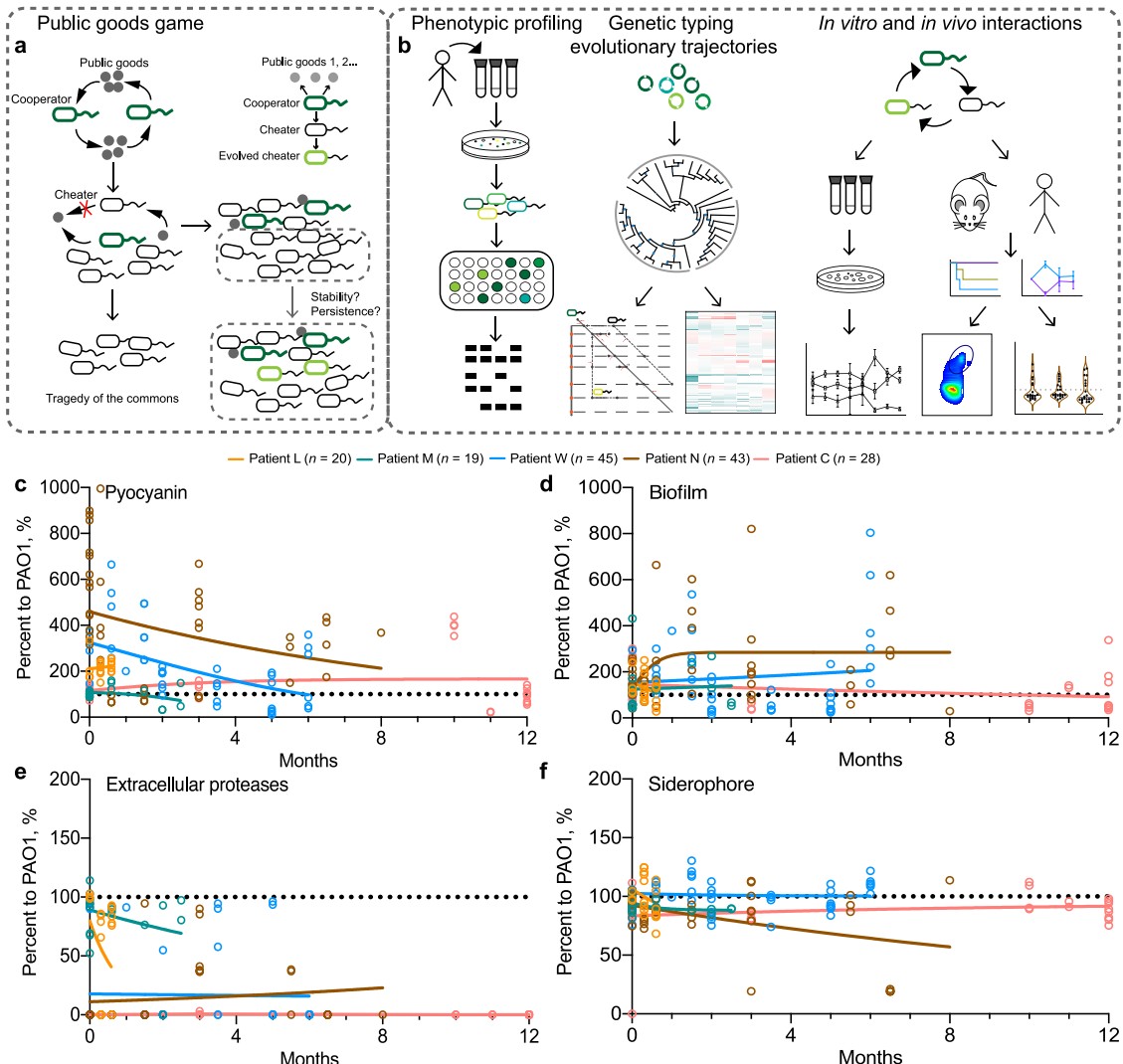

**Fig. 1 | Experimental design and phenotypic changes of *P. aeruginosa* isolates longitudinally collected from host lungs. a** Theoretical model of bacterial public goods game. Public good-producers (cooperators) can be invaded by the non-producers (cheaters), and the enrichment of nonproducers may cause the tragedy of the commons. We further hypothesize that the evolution of cheaters may lead to a transition of public goods game and facilitate the stabilization and persistent colonization of polymorphic population. **b** Experimental design. *P. aeruginosa* isolates were longitudinally collected from the respiratory samples of COPD patients, followed by characterizing their phenotypic and genetic features, phylogenetic status, evolutionary trajectories, and the outcomes of social interactions in vitro and in vivo. **c–f** Sampling period-dependent changes of the capacities of *P. aeruginosa* COPD isolates to produce extracellular products. Data shown are the phenotypic values of each isolate (symbols) normalized to those of the reference strain PAO1 and the variation tendency (trendlines) in each patient.

during the evolution of *P. aeruginosa* in COPD airways. The evolution of the polymorphic *P. aeruginosa* population was characterized by the evolvement of different *lasR* mutants, as determined by the evolutionary trajectory analysis. By exploring the interactions of different *lasR* mutants and the ancestral *P. aeruginosa*, we found that the structure of the polymorphic population could be stabilized by the presence of sequentially evolved *lasR* mutants under different conditions. Finally, the contribution of evolved *lasR*-mutant to the survival advantage of *P. aeruginosa* population was verified by using mouse models and clinical respiratory samples.

## Results

### In vivo evolution of *P. aeruginosa* leads to the sequential enrichment of isolates deficient in producing different sharable products

To explore the social traits of *P. aeruginosa* in host lungs, 25 *P. aeruginosa*-positive COPD patients (but negative in the past whole year) were enrolled for longitudinally collecting *P. aeruginosa*. BALs or spontaneous sputa of the patients were recovered at intervals of ~15 days, and a total of 536 *P. aeruginosa* colonies were obtained. The patients could be divided into three groups according to their lifetime after the first sampling: Group A included fourteen patients who passed away within 0.5 to 2.5 months because of severe lung infection, Group B included three patients who lived for 4–8 months, and Group C included only one patient who became *P. aeruginosa* negative after 12 months. The remaining seven patients quit the project after 1 or 2 sampling periods because of remission and other uncontrollable factors. *P. aeruginosa* isolates of patient L (80 years old) and patient M (74 years old), who were randomly selected from Group A to represent the patients with aucte lung infection, patient W (92 years old) and patient N (81 years old), who represented the patients with chronic lung infection in Group B and received a relatively longer time of sampling, and the sole patient C (66 years old) in Group C, were selected for further analyses.

Among the virulence-related phenotypes of *P. aeruginosa*, we mainly evaluated the levels of extracellular products (such as pyocyanin, biofilm, extracellular proteases, and siderophores) produced by the isolates from each patient. The results showed that the majority of isolates from each patient throughout the sampling periods produced higher or comparable levels of pyocyanin and biofilms to the reference strain *P. aeruginosa* PAO1. By contrast, the production of extracellular proteases and siderophores, especially the former by the COPD isolates, was lower than those of PAO1 and gradually reduced in patients L and M over time, while it remained on the low side by the isolates of the other three patients (Fig. 1c–f). Moreover, all the *P. aeruginosa* isolates of patient C were deficient in producing QS-controlled extracellular proteases, and this patient was still alive after the sampling was finished. We then sequenced the *lasR* (encodes the central QS regulator) and *pvdS* (regulates the synthesis of siderophore pyoverdine) genes of the *P. aeruginosa* isolates (*n* = 145) from the five COPD patients. The results showed that the emergence of *lasR* mutants was earlier than that of *pvdS* mutants, and about one sixth of the *lasR* mutants from patient W co-carried a mutation in the *pvdS* gene (Fig. S1). Therefore, these data revealed that the *P. aeruginosa* isolates deficient in producing the costly and sharable extracellular products were selected in a sequential way. *P. aeruginosa lasR* mutants deficient in producing extracellular proteases might be selected first in COPD airways, while the *lasR* mutants carrying additional mutations in the *pvdS* gene might be selected during further evolution.

### COPD airways are frequently colonized by polymorphic *P. aeruginosa* population

The genetic relationship of the *P. aeruginosa* COPD isolates were preliminarly sorted by enterobacterial repetitive intergenic consensus-polymerase chain reaction (ERIC-PCR)-based isolate typing. The

results revealed that the isolates longitudinally collected from the same COPD patient could be generally separated into two or three subgroups (Fig. S2), indicating the colonization of multiclonal *P. aeruginosa*. The isolates from each sampling period of different patients and representing the phenotypes of others in each subclade of the isolate typing were selected for whole-genome sequencing (WGS)-based comparative genomic analyses.

The genome sizes of the sequenced *P. aeruginosa* COPD isolates ranged from 6.19 to 6.74 Mbp (6.49 ± 0.116 Mbp), with an average GC content of 66.38% and gene number of 5977 ± 121.20 (Dataset S1). Single nucleotide polymorphism (SNP)-based phylogenetic analysis revealed that these isolates could be separated into three main clades by also considering their multilocus sequence typing (MLST) and serotyping results (Fig. 2a). Specifically, Clade 1 was composed of 13 *P. aeruginosa* isolates belonging to MLST type ST357 and serotype O11 (ST357:O11) obtained from almost all sampling periods of patient W. The isolates in Clade 1 showed a close relationship with the hypervirulent clinical isolate *P. aeruginosa* PA14. Clade 2 was composed of all the isolates from patient M, three isolates from patient W, and isolate C1a from patient C. These isolates belonged to the same clone type of ST233:O6. The composition of the *P. aeruginosa* isolates in Clade 3 was complex in terms of their isolation source and clone type, including all the isolates from patient L (ST549:O5 and ST181:O3), four isolates from patient W (ST1129:O6 and ST549:O3), and isolate C5a from patient C (ST274:O3). These isolates were clustered with the model *P. aeruginosa* isolates PAK, PAO1, and CF isolate DK2, and separated from the PA14 group (Clade 1) with high supporting rate (Fig. 2a). Therefore, these results clearly demonstrated the cross-transmission of *P. aeruginosa* among patients and confirmed that COPD airways were frequently cocolonized by polymorphic *P. aeruginosa*.

### *lasR* mutants are sequentially evolved in the polymorphic *P. aeruginosa* population

We then set out to study the evolutionary dynamics of the polymorphic *P. aeruginosa* population in lung environments and the genetic features that were associated with the differentiation of social structure. The isolates from patient W were used to predict the in vivo evolutionary trajectories of *P. aeruginosa*, because they showed an abundant genetic diversity and located in the position close to the main branch of each phylogenetic group (Fig. 2a). As shown in Fig. 2b, W1a was identified as the primary colonizing clone because it was initially isolated from the patient and produced comparable levels of extracellular products to PAO1. All the isolates with the same clone type of ST357:O11 distributed in the Clade 1 of the phylogenetic tree were defined as the main evolutionary group (Fig. 2a, b). W1c, W4a, and W6b, with the clone type of ST1129:O6, clustered together in Clade 3 of the phylogenetic tree and were defined as Subgroup 1. Finally, isolates W3a, W5b, and W7f, with the clone type of ST233:O6, clustered together in Clade 2 of the phylogenetic tree were defined as Subgroup 2.

Compared to the ancestral isolate W1a, all the core isolates in the main group harbored 5 nonsynonymous SNPs in *lasR* and *dctM* (C4-dicarboxylate transporter) and 3 non-shifted insertion/deletion sites (InDels) in *PAO987* and *coaA* (Table 1 and Figs. 2b and S3). Specifically, only W1b harbored an additional nonsynonymous SNP in the type VI secretion system (T6SS) effector-encoding gene *tse1*. W2a harbored 1 nonsynonymous and 1 premature stop SNP in *tse1* and 1 non-shifted InDel in *xqhA* (T2SS secretion protein). The premature stop SNP in *tse1* of W2a was passed on to W2c. W2c lost the parental non-shifted InDel in *PAO987* but harbored a new frame-shifted InDel in *PAO979*. W4c acquired an additional shifted InDel in *PAO979* compared to W4b. W5c gained 7 SNPs, including 1 premature stop and 6 nonsynonymous SNPs in *dctM*, *PAO946*, and *PA4900*, and lost 1 non-shifted InDel site in *coaA* and 2 nonsynonymous SNPs in *dctM*. W6a lost the non-shifted InDel in *PAO987* and harbored 1 non-shifted InDel in *fha1* (T6SS secretion protein) and 3 new SNPs. Compared to other core isolates, 1 premature

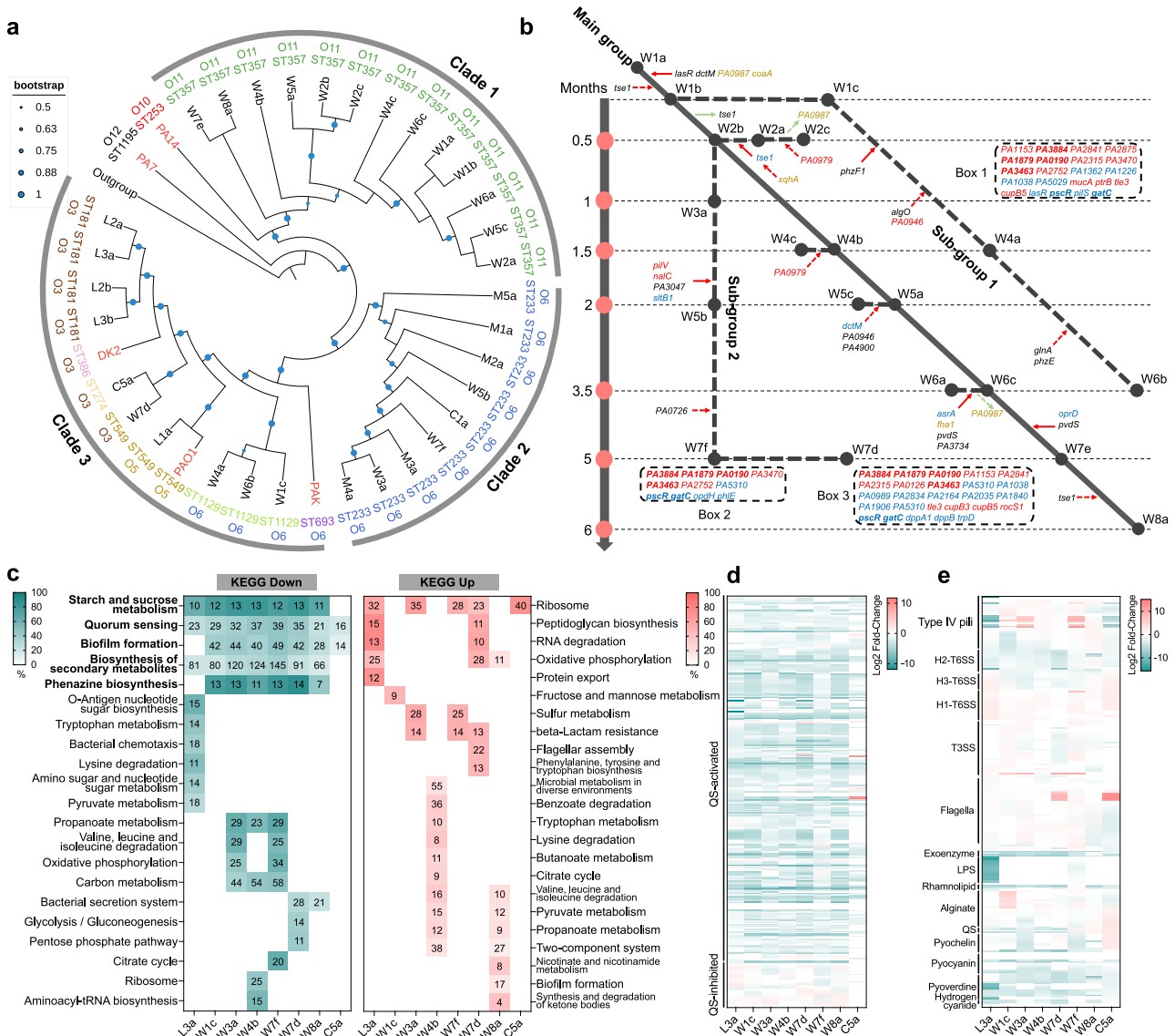

**Fig. 2 | Evolution of *P. aeruginosa* isolates in host lungs. a** SNP-based phylogenetic tree of *P. aeruginosa* COPD isolates constructed using maximum likelihood method. The size of blue dot at each node indicates bootstrap value (1000 replicates). All the isolates can be generally separated into three groups. The colored outer ring indicates MLST and serotypes. Outgroup, *Azotobacter vinelandii*. **b** Evolutionary trajectories of *P. aeruginosa* isolates in patient W. Time-dependent accumulation of mutation sites in the genes of *P. aeruginosa* isolates from patient W were determined by comparing the whole-genome sequence of each isolate to W1a. Red solid arrow, acquired fixed mutation. Red dotted arrow, acquired accidental mutation. Green solid arrow, lost fixed mutation. Green dotted arrow, lost accidental mutation. Blue color, gene with high effect SNP. Red color, gene with frameshifted InDel. Black color, gene with nonsynonymous SNP. Yellow color, gene with

non-shifted InDel. Bold, mutated genes shared by the subgroups. Box 1, Box 2, and Box 3 show the genes with loss-of-function mutations harbored by the isolates of subgroups 1, 2 and W7d compared to W1c, respectively. **c** KEGG terms enriched by the significantly down- and upregulated genes of evolved *P. aeruginosa* COPD isolates compared to the corresponding initial isolates (*Padj* < 0.05, automatically calculated by KOBAS v2.0). Heatmap color indicates percentage of differentially express gene number (black number in the colored squares) to the background gene number in each KEGG term. **d** Expression changes of QS-controlled genes (315 QS-activated and 38 QS-inhibited) in evolved *P. aeruginosa* COPD isolates compared to the corresponding initial isolates. **e** Expression changes of virulence-related genes in evolved *P. aeruginosa* COPD isolates compared to the corresponding initial isolates.

stop SNP in *oprD* (multifunctional outer membrane porin) and 1 nonsynonymous SNP in *pvdS* (pyoverdine biosynthesis regulatory gene) were fixed in W7e and W8a, and W8a accumulated another nonsynonymous SNP in *tse1*. In Subgroup 1, W1c, W4a, and W6b carried premature stop SNP sites in *lasR* and had the same categories of mutated genes with loss-of-function mutations compared to W1a (Table 1 and Fig. 2b-Box 1). In comparison to W1c, W4a harbored new nonsynonymous SNPs in *phzF1* and *algO* and a frame-shifted InDel in *PA0946*. The nonsynonymous SNP in *phzF1* of W4b was passed on to W6b. W6b also harbored new nonsynonymous SNPs in *glnA* and *phzE*. In Subgroup 2, W3a, W5b, and W7f had the same categories of genes

harboring loss-of-function mutations (Table 1 and Fig. 2b-Box 2), and the majority of the common genes were detected in W7d from another lineage (Table 1 and Fig. 2b-Box 3). In comparison to W3a, W5b harbored new nonsynonymous SNP in *PA3047*, premature stop SNP in *sltB1*, and frame-shifted InDels in *pilV* and *nalC*. These mutations in W5b were passed on to W7f. W7f also harbored a new nonsynonymous SNP in *PA0726*. These results indicated that the evolution of polymorphic *P. aeruginosa* population in COPD airways might lead to the sequential evolution of *lasR* mutants with potential functional changes, especially in the secretion and regulation of extracellular products.

**Table 1 | Mutation genes and types in *P. aeruginosa* COPD isolates from patient W compared to the initial isolate W1a**

| Isolates | Nonsyn._SNP | Nonsyno._Start | Nonsyn._Stop | Prem._Stop | Frame shifted InDels |
|---|---|---|---|---|---|
| W1b | *lasR, dctM, tse1* | – | – | – | – |
| W2a | *lasR, dctM, tse1* | – | – | *tse1* | – |
| W2b | *lasR, dctM* | – | – | – | – |
| W2c | *lasR, dctM, tse1* | – | *tse1* | *tse1* | *PA0979* |
| W4b | *lasR, dctM* | – | – | – | – |
| W4c | *lasR, dctM* | – | – | – | *PA0979* |
| W5a | *lasR, dctM* | – | – | – | – |
| W5c | *lasR, dctM, PA4900, PA0946* | – | *dctM* | – | – |
| W6a | *lasR, dctM, PA3734, pvdS* | – | *asrA* | – | – |
| W6c | *lasR, dctM* | – | – | – | – |
| W7e | *lasR, dctM, pvdS* | – | – – | *oprD* | – |
| W8a | *lasR, dctM, tse1, pvdS* | – | – | *oprD* | – |
| W1c | 3336 genes | *ynfL, PA1038, PA1226, PA1362* | *gatC* | *lasR, pscR, pilS* | *tle3, mucA, ptrB, cupB5, yheU, yqaA, PA0190, PA1153, PA1879, PA2315, PA2841, PA2875, PA3470, PA3884* |
| W4a[a] | *phzF1, algO, PA0946* | – | – | – | *PA0946* |
| W6b[a] | *phzF1, phzE, glnA* | – | – | – | – |
| W3a | 3280 genes | *opdH, phlE, ymdC* | *gatC* | *pscR* | *yheU, yqaA, PA0190, PA1879, PA3470, PA3884* |
| W5b[b] | *PA3047* | – | – | *sltB1* | *pilV, nalC* |
| W7f[b] | *PA3047, PA0726* | *PA1038, ymdC, PA0989, PA2834, dppA1, dppB, PA2164, PA2035* | – | *sltB1, trpD* | *pilV, nalC* |
| W7d | 3354 genes | *ynfL, PA1038, ymdC* | *gatC, bkdR* | *pscR, algR* | *tle3, cupB3, rocS1, vfr, yheU, yqaAPA0126, PA0190, PA1153, PA1879, PA2315, PA2841, PA3884* |

[a]The genome sequences of the isolates belonging to Subgroup 1 in Fig. 2b were compared to that of W1c.
[b] The genome sequences of the isolates belonging to Subgroup 2 in Fig. 2b were compared to that of W3a. Nonsyn._SNP, Nonsynonymous SNP. Nonsyno._Start, Nonsynonymous SNP happened in the start codon. Nonsyno._Stop, Nonsynonymous SNP happened in the start codon. Prem._Stop, Nonsynonymous SNP resulted in a premature stop codon. '—', Not detected.

To investigate the effect of mutations on the functional categories of *P. aeruginosa*, the global transcriptions of *P. aeruginosa* isolates from the initial and final sampling periods and those from the subgroups of patient W were profiled by RNA sequencing (RNA-seq). Compared to the corresponding initial isolates, the results of KEGG pathway prediction demonstrated the greatest decreases in functional categories related to QS, followed by starch and sucrose metabolism, biofilm formation, biosynthesis of secondary metabolites, and phenazine biosynthesis in all the evolved *P. aeruginosa* isolates (Fig. 2c and Dataset S2). Interestingly, the 5 commonly decreased KEGG terms were the same as those enriched by the genes activated by the two main QS regulatory genes, *lasR* and *rhlR* (Dataset S2). When the significantly differentially expressed genes were mapped to the list of QS-controlled (315 activated and 38 inhibited) genes[26], we found that all the isolates showed decreased expression of a large number of genes activated by QS (Figs. 2d and S4). Additionally, the significantly enriched KEGG terms differed among the evolved isolates as the sampling time increased (Fig. 2c). We further compared the expression levels of all the virulence genes (according to the Virulence Factor Database) in the evolved *P. aeruginosa* to the corresponding initial isolates. The results showed that the expression levels of genes related to QS-controlled extracellular products, iron acquisition, H2- and H3-T6SS were decreased, while the genes related to H1-T6SS and flagella were generally increased. On the other hand, the expression levels of genes related to type IV pili, T3SS, and alginate production varied among isolates (Fig. 3e and Dataset S3). Therefore, these results combined with the generally decreased production of QS-controlled extracellular proteases by *P. aeruginosa* isolates from COPD airways (Fig. 1c–f), revealed that the transcriptional divergence of QS regulation caused by the mutation of *lasR* gene commonly occurred in the polymorphic *P. aeruginosa* population, while the accumulation of other mutations might be associated with the adaptability of *lasR* mutants during further evolution.

**Involvment of sequentially evolved *lasR* mutants in the polymorphic *P. aeruginosa* population stabilizes social structure**

Based on the genetic and phenotypic characteristics of *P. aeruginosa* isolates determined above (Table 1 and Figs. 2 and S1), the ancestral isolate W1a (herein renamed as W1a-ST357), which belonged to the lineage ST357:O11, the co-isolated W1c (herein renamed as W1c-ST1129-*Exp⁻Pvd⁻*), which belonged to the lineage ST1129:O6 with several loss-of-function mutations in pathoadaptive genes and was deficient in producing extracellular protease (*Exp⁻*) and pyoverdine (*Pvd⁻*), and the sequentially identified isolates W4b (herein renamed as W4b-ST357-*Exp⁻*) with shifted InDel mutation in *lasR* and deficient extracellular protease-producing ability and W8a (herein renamed as W8a-ST357-*Exp⁻Pvd⁻*) with an additional shifted InDel site in *pvdS* and deficient abilities in producing extracellular protease and pyoverdine, were selected to study their interactions in the scenario of public goods game. We first showed that when the isolates were monocultured in M9 casamino acids (CAA, hydrolysates of casein), which do not required QS for bacterial growth, W4b-ST357-*Exp⁻* and W8a-ST357-*Exp⁻Pvd⁻* grew slightly faster than W1a-ST357 in the initial 12 h but became slower after 20 h, and W1c-ST1129-*Exp⁻Pvd⁻* also grew slower than the other isolates (Fig. S5a). By contrast, the growth of W4b-ST357-*Exp⁻* and W8a-ST357-*Exp⁻Pvd⁻* in the QS-required medium (M9-casein) was slower than that of W1a-ST357 in the initial 6 h and then became comparable to W1a-ST357, while the growth of W1c-ST1129-*Exp⁻Pvd⁻* was remarkably slower than the other isolates (Fig. S5b). Correspondingly, W1a-ST357 produced a large proteolytic ring on M9-casein plates, W4b-ST357-*Exp⁻* and W8a-ST357-*Exp⁻Pvd⁻* produced smaller proteolytic rings, while the capacity of W1a-ST1129-*Exp⁻Pvd⁻* to

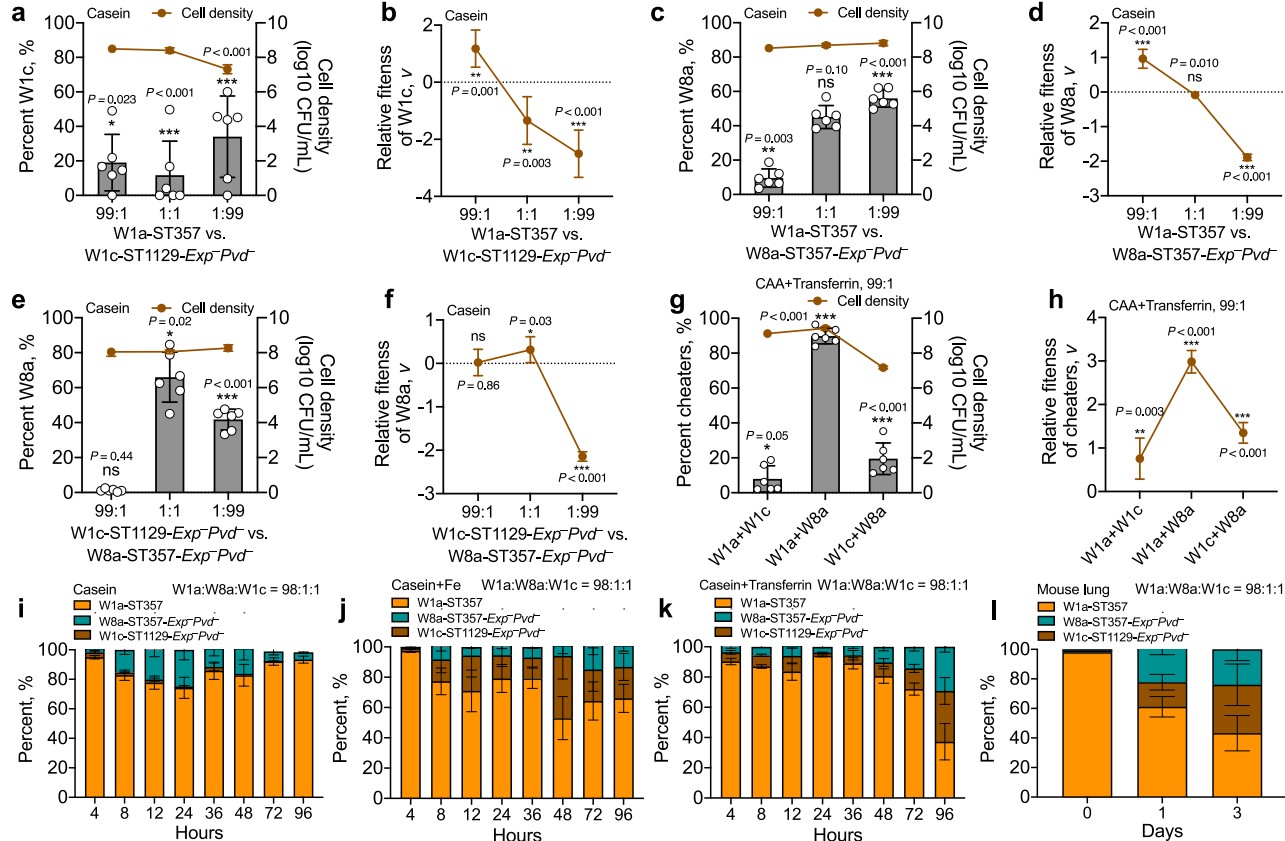

**Fig. 3 | In vitro and in vivo competitions of *P. aeruginosa* COPD isolates under different conditions.** *P. aeruginosa* isolates W1a-ST357, W1c-ST1129-*Exp⁻Pvd⁻*, and W8a-ST357-*Exp⁻Pvd⁻* were cocultured in double or triple in 2 mL of M9 minimal growth medium supplemented with different carbon sources and iron levels from different initial ratios. Frequencies and relative fitness of W1c-ST1129-*Exp⁻Pvd⁻* or W8a-ST357-*Exp⁻Pvd⁻* in double cocultures of (**a, b**) W1a-ST357 + W1c-ST1129-*Exp⁻Pvd⁻*, (**c, d**) W1a-ST357 + W8a-ST357-*Exp⁻Pvd⁻*, or (**e, f**) W1c-ST1129-*Exp⁻Pvd⁻* + W8a-ST357-*Exp⁻Pvd⁻* in M9-casein medium for 24 h.
**g** Frequencies and (**h**) relative fitness of the isolates with an initial frequency of 1% in double cocultures of W1a-ST357 + W1c-ST1129-*Exp⁻Pvd⁻*, W1a-ST357 + W8a-ST357-

*Exp⁻Pvd⁻*, or W1c-ST1129-*Exp⁻Pvd⁻*+W8a-ST357-*Exp⁻Pvd⁻* in iron-limiting M9-CAA medium for 24 h. Right Y-axis indicates the cell densities of each culture. The value of each column was compared to the initial frequency of corresponding isolate using two-tailed unpaired *t*-test. *P < 0.05. **P < 0.01. ***P < 0.001. Dynamic changes in the frequencies of W1a-ST357, W1c-ST1129-*Exp⁻Pvd⁻*, and W8a-ST357-*Exp⁻Pvd⁻* during the coevolution of the three isolates in (**i**) M9-casein, (**j**) iron-abundant M9-casein, (**k**) iron-limiting M9-casein media, and (**l**) mouse lungs from an initial ratio of 98:1:1. The culture media were refreshed at 24 h interval and the experiment was stopped when the frequency of each isolate in the culture was relatively stable. Data shown are means ± standard deviation (SD) of six independent replicates.

produce the QS-controlled extracellular proteases was completely abolished (Fig. S5c).

We then performed a batch of competition assays to investigate the social interactions of *P. aeruginosa* isolates by coculturing different combinations of them in M9-casein medium. The results showed that both the growth of W1c-ST1129-*Exp⁻Pvd⁻* and W8a-ST357-*Exp⁻Pvd⁻* from a small initial frequency (1%) was faster than that of cocultured W1a-ST357 in the two-player game (Fig. 3a–d), indicative of the exploitation of W1a-ST357 by W1c-ST1129-*Exp⁻Pvd⁻* and W8a-ST357-*Exp⁻Pvd⁻*. W4b-ST357-*Exp⁻* as a transitional isolate from W1a-ST357 to W8a-ST357-*Exp⁻Pvd⁻* could also invade W1a-ST357 (Fig. S6). However, the three *lasR* mutants showed different degrees of exploitation on W1a-ST357, and the ability of W1a-ST357 to invade the three *lasR* mutants was also different. These differences might be associated with the different capacities of the *lasR* mutants to produce elastase (public goods) and the innate slow growth rate of W1c-ST1129-*Exp⁻Pvd⁻* (Fig. S5). We also tested the interaction of W1c-ST1129-*Exp⁻Pvd⁻* and W8a-ST357-*Exp⁻Pvd⁻* and found that, W8a-ST357-*Exp⁻Pvd⁻* failed to invade W1c-ST1129-*Exp⁻Pvd⁻* but could be readily exploited by W1c-ST1129-*Exp⁻Pvd⁻* (Fig. 3e, f). This might be due to the weak ability of W8a-ST357-*Exp⁻Pvd⁻* to produce the costly public goods, while W1c-ST1129-*Exp⁻Pvd⁻* produced nothing (Fig. S5c). Moreover, we found that W8a-ST357-*Exp⁻Pvd⁻* failed to invade its transitional parental isolate W4b-

ST357-*Exp⁻* from a small initial frequency in M9-CAA medium, but could significantly exploit W4b-ST357-*Exp⁻* under iron-depleted conditions that caused by the supplementation of Transferrin (Fig. S7). This result indicated that as two kinds of *lasR* mutants, the latterly identified isolate W8a-ST357-*Exp⁻Pvd⁻* evolved an additional capacity to invade W4b-ST357-*Exp⁻* in the competition for iron-chelating siderophores. W8a-ST357-*Exp⁻Pvd⁻* had a remarkably higher relative fitness than W1a-ST357 in the competition of siderophores and could also exploit W1c-ST1129-*Exp⁻Pvd⁻* (Fig. 3g, h). Therefore, our results demonstrated the cheating behaviors of differentially evolved *P. aeruginosa lasR* mutants from the same patient in playing public goods games in dependence on environmental conditions.

Because W4b-ST357-*Exp⁻* was a transitional isolate from W1a-ST357 to W8a-ST357-*Exp⁻Pvd⁻*, and the highly evolved isolate W1c-ST1129-*Exp⁻Pvd⁻* could also be invaded by W8a-ST357-*Exp⁻Pvd⁻* in the competition for iron (Fig. 3g, h), we further monitored the interaction dynamics of W1a-ST357, W8a-ST357-*Exp⁻Pvd⁻*, and W1c-ST1129-*Exp⁻Pvd⁻* by successively coculturing them (98:1:1) in M9-casein medium with different levels of iron. The results showed that in casein medium which creates the scenario of an extracellular protease-mediated public goods game, the frequency of W8a-ST357-*Exp⁻Pvd⁻* increased to 25.03 (± 6.82) % in the initial 24 h and then gradually decreased to 6.54 (± 2.35) % after 72 h, while W1c-ST1129-*Exp⁻Pvd⁻*

showed a weak ability to invade W1a-ST357 (Fig. 3i). Interestingly, the addition of iron in M9-casein medium resulted in the success of W1c-ST1129-*Exp⁻Pvd⁻*. This might be partially related to the restored growth of W1c-ST1129-*Exp⁻Pvd⁻* by supplementation with iron (Fig. S8). Both W8a-ST357-*Exp⁻Pvd⁻* and W1c-ST1129-*Exp⁻Pvd⁻* could invade W1a-ST357 and the three isolates coexisted with a ratio of approximately 3:1:1 (Fig. 3j). In casein+Transferrin medium, which creates the scenario of multiple public goods games, the three isolates could still coexist in the population with a W1a/W8a/W1c ratio of approximately 1:1:1 (Fig. 3k). Additionally, none of the three sets of successive sub-culturing assays incurred a collapse of the population, even the growth of the mixed population was relatively slow in casein+Transferrin medium (Fig. S9a).

We then tested the interactions of the three isolates in mouse lungs by chronically coinfecting them from an initial ration of 98:1:1. The results showed that W8a-ST357-*Exp⁻Pvd⁻* and W1c-ST1129-*Exp⁻Pvd⁻* could invade W1a-ST357 and coexist in lung environments by forming a ratio of approximately 1:1:1 (Figs. 3l and S9b). Moreover, the interactions of *P. aeruginosa* isolates from patient W were validated by repeating the experiments above (Fig. 3) using the isolates N1a (intact *lasR* and *pvdS* genes), N5c-*lasR* (*lasR* mutant), and N7d-*lasRpvdS* (*lasR* and *pvdS* double mutant) from patient N (Figs. S10 and S11). We further tested the competitions of *P. aeruginosa* isolates from patient W in artificial sputum medium (ASM). The results showed that although the monocultured W1a-ST357, W4b-ST357-*Exp⁻*, W8a-ST357-*Exp⁻Pvd⁻*, and W1c-ST1129-*Exp⁻Pvd⁻* showed similar growth status in ASM, W4b-ST357-*Exp⁻*, W8a-ST357-*Exp⁻Pvd⁻*, and W1c-ST1129-*Exp⁻Pvd⁻* could also invade W1a-ST357 under different coculture conditions from a low initial frequency (1%) (Figs. S12–S14). Therefore, these results collectively suggested that the sequentially evolved *lasR* mutants with varying capacities of producing the sharable extracellular products can interact with WT *P. aeruginosa* in a framework termed cascaded public goods game, which describes the potential transformation of bacterial social interaction from extracellular protease-mediated public goods game to siderophore-mediated, and thus contributes to the maintenance of cooperation and the relative stability of population structure in diverse environments.

## A mixture of ancestral *P. aeruginosa* and evolved *lasR* mutant compromises the host inflammatory responses

Based on the important role of *lasR* mutants during the evolution of the polymorphic *P. aeruginosa* population identified above (Figs. 2 and 3), we then tested the influence of the invasion of evolved *lasR* mutant on the adaptability (in terms of bacterial virulence and host immune fluctuation) of *P. aeruginosa* population by establishing chronic infection models using W1a-ST357, W1c-ST1129-*Exp⁻Pvd⁻*, and a mixture thereof (1:1). In the slow-killing assay (mimicking the chronic infection status of *P. aeruginosa*) conducted in the *Caenorhabditis elegans* infection model, W1a caused earlier death of the nematodes, but the results did not significantly differ from the mortality of nematodes infected by the mixture of W1a-ST357 and W1c-ST1129-*Exp⁻Pvd⁻* ($P = 0.1612$). In contrast, the evolved *lasR* mutant W1c-ST1129-*Exp⁻Pvd⁻* failed to kill any nematodes over 144 h (Fig. S15). Unsurprisingly, compared to the 100% survival rate of mice chronically challenged by W1c-ST1129-*Exp⁻Pvd⁻*, W1a-ST357 killed 80% of the mice, while W1a-ST357 and W1c-ST1129-*Exp⁻Pvd⁻* coinfection killed 50% in the initial 4 days (Fig. 4a). Moreover, although the growth of W1c-ST1129-*Exp⁻Pvd⁻* was remarkably slower than that of W1a-ST357 in vitro, it could persist in the lungs of mice infected by the mixture of W1a-ST357 and W1c-ST1129-*Exp⁻Pvd⁻* and showed less fluctuation of residual cell numbers in W1c-ST1129-*Exp⁻Pvd⁻*-infected mice (Figs. 4b, Sa and S16). These results suggested that W1a-ST357 had a strong pathogenic ability, while W1c-ST1129-*Exp⁻Pvd⁻* lost most of the lethality but adapted to persistent colonization. Therefore, we suspected that a mixture of W1a-ST357 and W1c-ST1129-*Exp⁻Pvd⁻* might combine the

advantages of the two strains and facilitate the colonization of *P. aeruginosa* during chronic lung infection.

The results of histological staining revealed that the W1a-ST357 and W1c-ST1129-*Exp⁻Pvd⁻* coinfection group induced more infiltration of immune cells to the mouse lung at 3 days post infection compared to the groups of mono-infection (Fig. 4c). We further examined the change of immune status in lung tissues induced by different combinations of *P. aeruginosa* during chronic infection by flow cytometry. Accordingly, the W1a-ST357 and W1c-ST1129-*Exp⁻Pvd⁻* groups recruited higher levels of inflammatory macrophages, neutrophils, and CD8⁺ T cells compared to those of the control group, while the coinfection group recruited relatively fewer immune cells with no significant differences (Figs. 4d–f and S17). However, the lung tissues had the lowest level of CD4⁺ T cells in the W1a-ST357 group, and there were no significant differences in CD4⁺ T cells between the coinfection and W1a-ST357 and W1c-ST1129-*Exp⁻Pvd⁻* groups compared to the counterparts of the uninfected control group. This indicated that the W1a-ST357 and W1c-ST1129-*Exp⁻Pvd⁻* infected mice had the most significant immune responses on day 3 (Figs. 4g and S17c). Furthermore, the levels of immune cells in the lungs of these three groups on day 7 were similar to those comparts on day 3, except for CD4⁺ T cells, which were no longer significantly different among the different groups (Figs. 4h–k and S18).

## Evolved polymorphic *P. aeruginosa* population restrains the volatility of host inflammatory gene expression

To better understand the effects of the evolved *lasR* mutant included polymorphic *P. aeruginosa* population on host immune responses, RNA-seq was then performed using mouse lung tissues above (Fig. 4b). At day 3 post infection, all the three infection groups triggered significant transcriptional changes in mouse lungs compared to the uninfected group. The most enriched upregulated signaling pathways were cytokine-cytokine receptor interaction, IL-17, TNF, NF-κB, and Nod-like receptor signaling pathways, etc. (Fig. 5a). However, the fold-change of differently expressed genes was significantly different among the three groups. The W1c-ST1129-*Exp⁻Pvd⁻* group had the most significantly changed genes related to the host immune response, while the W1a-ST357 and W1c-ST1129-*Exp⁻Pvd⁻* coinfection group had the fewest (Fig. 5a). We further divided these significantly changed genes into three categories: immunomodulatory molecules, cytokine-receptor interaction, and ECM (extracellular matrix)-receptor interaction. The expression of many representative molecules involved in host immune defense processes, such as Mevf, IL23a, and Mmp9, had the smallest fold-change in the W1a-ST357 and W1c-ST1129-*Exp⁻Pvd⁻* coinfection group (Figs. 5b–d and S19a–c). Consistently, the coinfection group also induced the slightest dynamic transcriptional change in mice compared with the other two groups between day 3 and day 7 (Figs. 5e and S19d). These results indicated that the polymorphic *P. aeruginosa* population comprised of *lasR*-intact and evolved *lasR* mutant individuals could minimize the immune fluctuation for persistent colonization in host lungs.

We then validated the immune responses of the hosts by profiling the proteomes of BALs from different sampling periods of patient L (colonized by more QS-intact *P. aeruginosa*), patient C (colonized by QS-deficient *P. aeruginosa*), and patient W (colonized by polymorphic *P. aeruginosa*). The results of mass spectrometry showed that although the BALs from the third sampling period of patients L and W had more significantly expressed proteins compared to the initial sampling period, the immune-related proteins in patient W, who was colonized by polymorphic *P. aeruginosa* population until death showed the smallest expression fluctuation (Figs. 5f, g, and S19e). We further identified that the expression levels of immune-related proteins in the BALs from different sampling periods of patient W remained relatively stable (Fig. 5h). We finally evaluated the transcriptional stability of the levels of immune-related genes in the BALs from different sampling

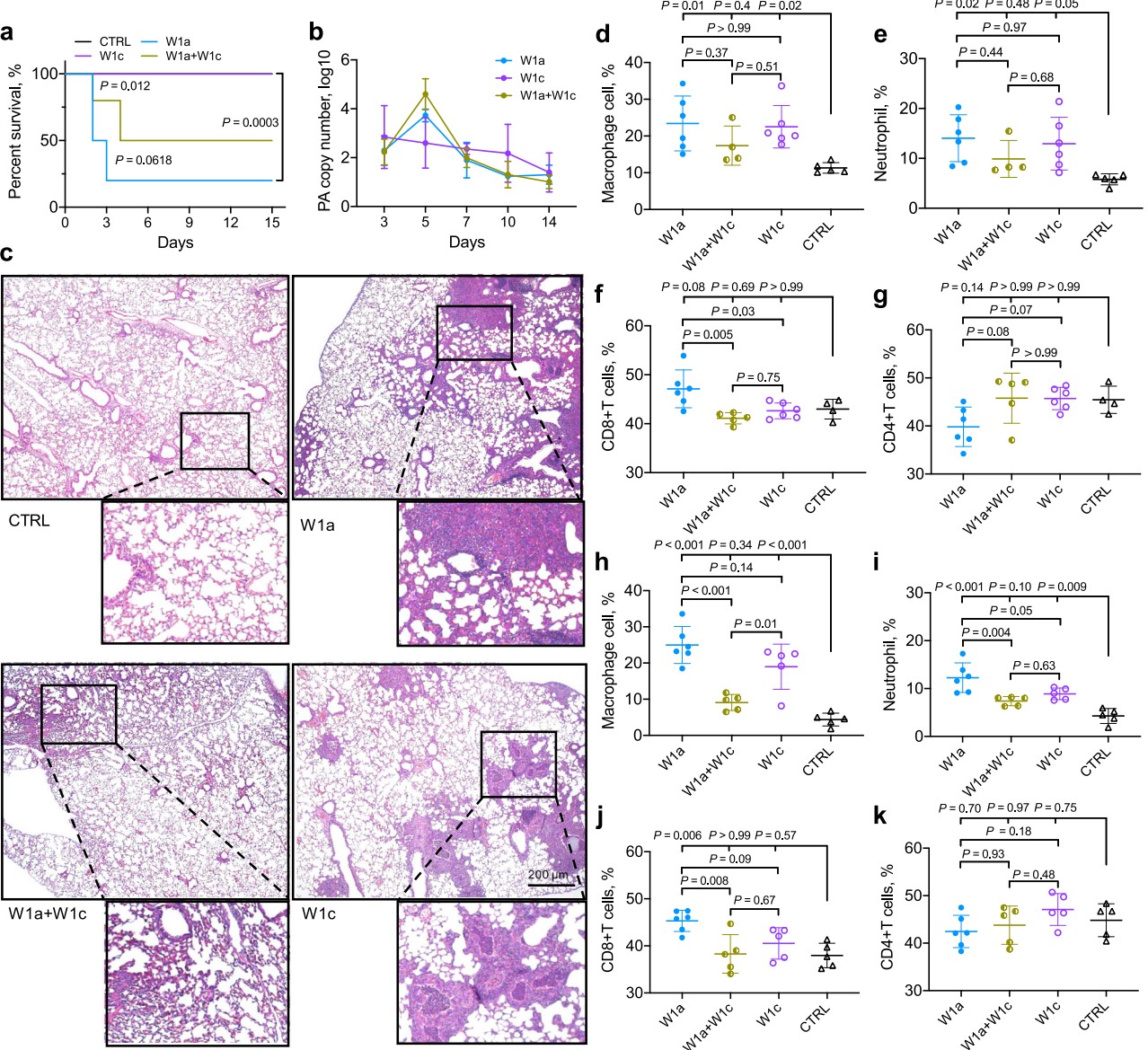

**Fig. 4 | Pathogenicity of *lasR*-intact *P. aeruginosa* and evolved *lasR* mutant in mouse model.** **a** Chronic lung infection mouse model (40 mice per group) using *P. aeruginosa* COPD isolates W1a-ST357, W1c-ST1129-*Exp⁻Pvd⁻*, and a 1:1 mixture of them. The survival curves of mice were compared by using Log-rank (Mantel−Cox) test. CTRL indicates the control group of mice without *P. aeruginosa* challenge. The black line that indicates the CTRL group in panel (**a**) was by the purple line (W1c-ST1129-*Exp⁻Pvd⁻*infected group). **b** 16 S rRNA-based copy number change of *P. aeruginosa* during chronic infection of mouse lungs (*n* = 3). **c**–**k** Lung tissues of mice chronically infected with/without *P. aeruginosa* at (**c**−**g**) day 3 and (**h**−**k**) day 7 were collected. **c** Lungs embedded in formalin were evaluated by H&E staining. Images are representative of three independent replicates. Scale bar, 200 μm. **d**–**k** The proportions of inflammatory macrophages, neutrophils, CD8⁺ T cells, and CD4⁺ T cells were evaluated by flow cytometry. 4–6 mice per group. The data shown are the means ± SD of three independent replicates. Statistical significance was calculated using one-way ANOVA with Tukey post-hoc tests using a 95% confidence interval.

periods of patient N, who was also colonized by a mixture of QS-intact and QS-deficient *P. aeruginosa*, by using RNA-seq. Indeed, the expression levels of the majority of immune-related proteins in the BALs from the fifth and eighth sampling periods were comparable to those from the first period (Fig. 5i). Altogether, these results suggested that the genetic evolution of QS mutants during chronic infection in the population greatly influences the immune status of the hosts.

## Discussion

The exertion and evolution of group behaviors in bacterial populations greatly expand the habitats and enhance the survival fitness of these microorganisms. The successful connection of public goods game and bacterial interaction has remarkably facilitated studies focusing on the persistent colonization and pathogenesis of bacteria in the theoretical context of sociomicrobiology[1,4,27]. In the present study, we characterize the evolutionary dynamics of the polymorphic *P. aeruginosa* population in COPD airways and identify a cascaded public goods game mediated by WT *P. aeruginosa* and continuously evolving *lasR* mutants that stabilizes social cooperation. Moreover, we find that the involvement of evolved *lasR* mutants in *P. aeruginosa* population facilitate the development of chronic infection by restraining immune fluctuation.

*P. aeruginosa* harbors several social traits, especially QS which plays vital roles in nutrient acquisition, defense and virulence, and the mutants deficient in producing the sharable extracellular products are frequently selected during evolution[28–32]. In the vast majority of cases, *lasR* mutants would coexist with cooperators in the population and the

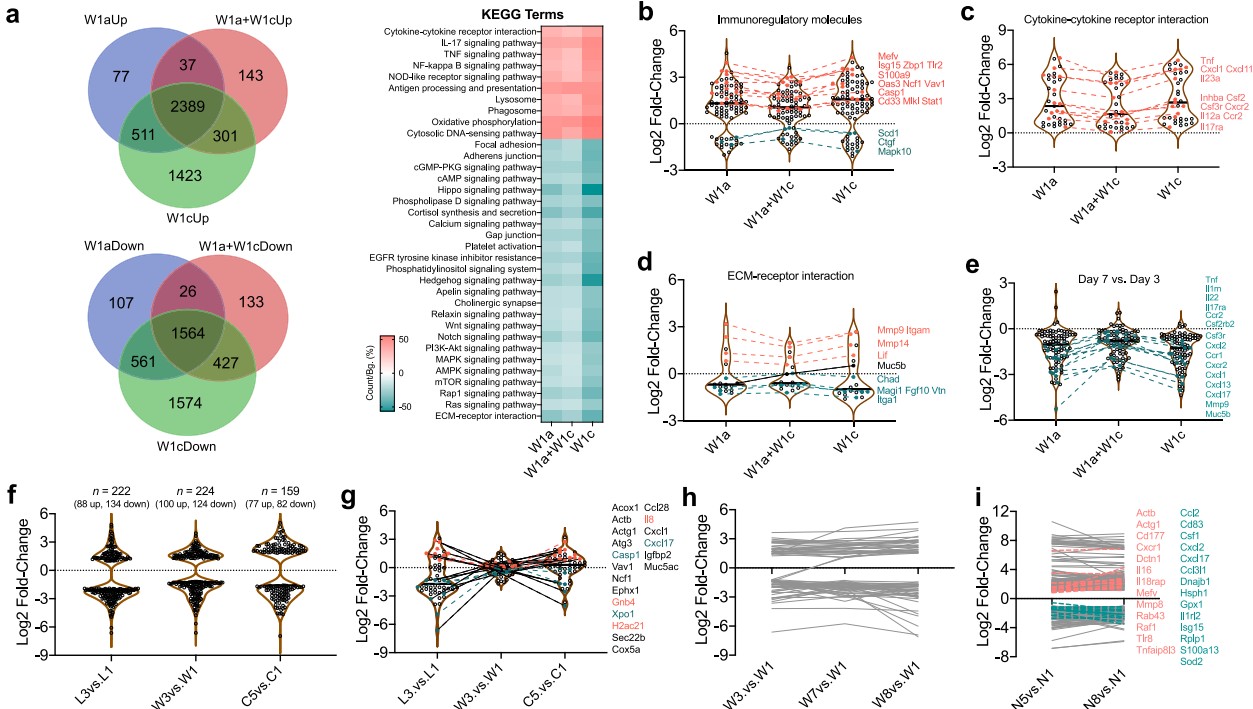

**Fig. 5 | Influence of polymorphic *P. aeruginosa* infection on the inflammatory responses of host lungs.** Lung tissues were collected from *P. aeruginosa* W1a-ST357 and W1c-ST1129-*Exp⁻Pvd⁻* chronically infected mice at day 3 for RNA-seq. **a** Venn diagram and KEGG enrichment analysis of the differentially expressed genes. **b**–**d** The significantly changed genes were divided into three categories: (**b**) immunomodulatory molecules, (**c**) cytokine-receptor interaction, and (**d**) ECM-receptor interaction. The fold changes of the differentially expressed genes between the three groups were compared. **e** The comparison of the differentially expressed genes between day 3 and day 7. The fold changes of the (**f**) total and (**g**) immune-related differentially expressed proteins between different sampling periods of patient L (the first and the last rounds), patient C (the first and the last rounds), and patient W (rounds 1, 3, 7, and 8). **h** The expression levels of immune-related proteins in the BALs from different sampling periods of patient W. **i** The expression changes of immune-related genes in the BALs from different sampling periods of patient N.

frequencies of them were dynamically changed. The present study reveals that COPD airways are susceptible to being cocolonized by *P. aeruginosa* with complex pathoadaptive phenotypes and mutations, typically the QS-intact and QS-deficient isolates, until the end of patients' lives (Fig. 1c–f and Table 1). These evidences indicate that coexistence of population with polymorphic genetic structure may be an optimal strategy for *P. aeruginosa* to persist in host tissues.

A recent work by Azimi et al. [33] studied the long-term evolution of monocultured *P. aeruginosa* PAO1 in synthetic sputum medium. They identified abundant mutant isolates in the early evolving population, while the enhanced β-lactam antibiotics of the polymorphic population during further evolution was positively correlated with *lasR* mutant frequency. However, it seems that the enhanced antibiotic resistance of the population was due to the enrichment of *lasR* mutant, rather than the selection of *lasR* mutant with sequentially evolved resistance-related mutation/s. Because the susceptibility of *lasR* mutant to β-lactam antibiotics could be restored by the complementation of intact *lasR* gene[33]. By characterizing the evolutionary trajectories of *P. aeruginosa* isolates from COPD airways, here we shows that the *lasR* mutants are successively evolved and play important roles during the evolution of the polymorphic population (Fig. 2b and Table 1). In the previously reported interaction of cheating on cheaters, WT *P. aeruginosa* could be exploited by its isogeneic *lasR* mutant in a QS-controlled product-mediated public goods game, and the *pvdS* mutant could invade the *lasR* mutant in a siderophore-mediated public goods game[24]. Our present study identifies a cascaded public goods game among the sequentially evolved *P. aeruginosa* isolates from COPD airways. All the three *lasR* mutants (W1c-ST1129-*Exp⁻Pvd⁻*, W4b-ST357-*Exp⁻*, and W8a-ST357-*Exp⁻Pvd⁻*) could invade W1a-ST357 in QS-required medium, and W8a-ST357-*Exp⁻Pvd⁻* with an

additional mutation in the *pvdS* gene also had a growth advantage over the other isolates in iron-limitation medium (Figs. 3a–h and S7). Moreover, cooperation were stabilized in the competitions of W1a-ST357, W1c-ST1129-*Exp⁻Pvd⁻*, and W8a-ST357-*Exp⁻Pvd⁻*, and the frequencies of which would be changed under different environmental conditions (Fig. 3j–l). These results indicate that the interaction of *P. aeruginosa* in the polymorphic population might be multifactorial and transformable. We did not verify the interactions of these clinical isolates by using the model *P. aeruginosa* strains here, because the additional mutations in W4b-ST357-*Exp⁻* and W8a-ST357-*Exp⁻Pvd⁻* are unlikely to influence the functions of QS system and siderophore production (Table 1). Alternatively, the generality of the interactions among the isolates from patient W was tested by using ASM medium and a batch of parallel experiments using the similar isolates from patient N (Figs. S10−S14). Further collection and identification of sequentially evolved *lasR* mutants from more COPD patients may contribute to proving the generality of bacterial cascaded public goods game identified in the present study.

Our in vitro competition assays clearly demonstrates the roles of different *lasR* mutants in the social interactions with QS-intact *P. aeruginosa*. However, it is hard to directly determine whether the enrichment of *lasR* mutants in host lungs was attributed to their cheating on QS-intact *P. aeruginosa*, or to their survival advantages in adaptation to the lung environment. In comparison to the main group *P. aeruginosa* isolates from patient W, the isolates of the two subgroups were similar to the previously reported CF-adapted *lasR* mutants, which were collected from the later stage of CF airways and carried several loss-of-function mutations in pathoadaptive genes[25]. Typically, the isolate W1c-ST1129-*Exp⁻Pvd⁻*, which might be transmitted from an unknown reservoir and was co-isolated with W1a-ST357,

harbored a large amount of loss-of-function mutations and was capable of infecting host lungs without the presence of QS-intact *P. aeruginosa* (Figs. 2a, b and 4a, b). These results highlight the notion that *P. aeruginosa lasR* mutants may undergo further adaptive genetic evolution during chronic infection and have a survival advantage over the less evolved isolates. On the other side, the clearance of *P. aeruginosa* isolates (pure extracellular protease-negative) from the lung of patient C after 12 months indicates that the persistent colonization of *lasR* mutants might require the WT strains. Therefore, the fitness advantage of *lasR* mutants in the lung environments might be simultaneously associated with their adaptive evolution and social interactions with QS-intact *P. aeruginosa*.

Previous studies suggested that the reduced virulence of *P. aeruginosa* caused by the invasion of *lasR* mutant might be associated with the decreased production of QS-controlled products at population level[33,34]. Differently, Mould et al. [35] identified a cross-feeding interaction between *lasR*-intact and *lasR* mutant *P. aeruginosa* by showing that, the *lasR*-intact individuals would produce citrate upon the stimulation of pyochelin produced by *lasR* mutant, and exclusively activate the *rhl*-QS system of *lasR* mutant to hyperproduce pyocyanin and biofilm. LaFayette et al. [25] reported that the evolved *lasR* mutants from CF airways could elicit hyperinflammatory responses. These findings combined with the identification of *lasR* mutants with altered QS regulation from CF airways[36], collectively indicate the evolutionary selection on *lasR* mutants that can better adapt to the lung environments. The present study investigates the survival advantage of evolved polymorphic *P. aeruginosa* population in host lungs by using *lasR*-intact *P. aeruginosa* and the evolved *lasR* mutant. In contrast to the clearance of *lasR* mutant of *P. aeruginosa* reference strain by the host[37], W1c-ST1129-*Exp⁻Pvd⁻* showed an avirulent phenotype but could successfully colonize mouse lungs (Fig. 4a, b). Importantly, a mixture of *lasR*-intact *P. aeruginosa* and evolved *lasR* mutant induced the minimum immune fluctuation in a mouse model compared to those chronically infected by a single genotype isolate, and a similar trend was also observed in patient samples (Figs. 4 and 5). These data suggest that the coinfection of *lasR*-intact *P. aeruginosa* and evolved *lasR* mutant can alleviate the host immune fluctuations or responses. This also indicates that reducing the intensity of the host inflammatory response is crucial for the colonization and long-term survival of pathogenic bacteria.

Collectively, this study dissects the evolution and intraspecific interaction of *P. aeruginosa* longitudinally collected from COPD airways. We find that the evolution of *P. aeruginosa* population in lung environments sequentially selects the isolates deficient in producing the costly and sharable extracellular products. Persistent colonization of the polymorphic *P. aeruginosa* population is associated with the accumulation of mutations in the *lasR* mutants, and leads to the cocolonization of isolates with diverse capacities in playing the public goods game. The population structure can be stabilized by the formation of a cascaded public goods game mediated by QS-controlled extracellular products and siderophores. These results demonstrate the multistage evolution and complex interaction of *P. aeruginosa* in adaptation to the host lungs, and provide plausible explanations for the maintenance of cooperation in the game of public goods and the recurrence of *P. aeruginosa*-related chronic infections.

Moreover, the identification of the *lasR* mutant-centered cascaded public goods game in the present study also raises a significant concern regarding the flourishing development of antivirulence drugs targeting a single core regulator of the QS system[38–40]. Even the *lasR* mutant may produce the QS-controlled products by invoking the regulation of MvfR (PqsR), a sublevel regulator of the QS system, or elicit other unexpected host immune responses by harboring pathoadaptive mutations during further evolution[41,42]. Therefore, the in vivo evolutionary trajectories and interaction dynamics of *P. aeruginosa* isolates identified in the present study, as well as the

characterization of host immune responses induced by different combinations of them, provide a vital basis for further understanding the pathogenesis of *P. aeruginosa* and the development of therapeutic strategies against pseudomonal infections.

## Methods
### Ethical statement
BALs and sputum samples were obtained from the COPD patients hospitalized in the affiliated hospital of Chengdu University (Chengdu, China). Written informed consents were received from the patients or their immediate family members. The study was approved by the Ethics Committee of the Affiliated Hospital of Chengdu University (PJ2020-021-03), and all methods were carried out in accordance with the guidelines and regulations of Chengdu University. Mouse models used in this study were bought from Beijing Huafukang Laboratory Animal Co., Ltd. (Beijing, China) and routinely housed in the specific-pathogen-free facility of the State Key Laboratory of Biotherapy, Sichuan University. Animal experiments were approved by the Ethics Committee of the State Key Laboratory of Biotherapy (2021559A) and carried out in compliance with institutional guidelines concerning animal use and care of Sichuan University.

### Sample collection, bacterial strains, and culture media
A total of 25 patients (56–92 years old) who were diagnosed as COPD and *P. aeruginosa*-positive (but negative in the past whole year) hospitalized in the Affiliated Hospital of Chengdu University were enrolled for longitudinal collection of *P. aeruginosa*. BALs of the patients were recovered with endoscopic surgery using electronic bronchoscope (Olympus BF-1TQ290) at an interval of 15 days. Spontaneous sputa in the morning were collected from the patients if the endoscopic surgery was unnecessary. A portion of the sample was cultured in lysogeny broth (LB) at 37 °C with shaking (220 rpm) for 2 days. The culture liquid was spread on LB plate and cultured at 37 °C overnight. The colonies with apparent differences in shape, color, size, and surface states were picked out for 16S rDNA-based species identification. The single colony of *P. aeruginosa* clinical isolates and the reference isolate *P. aeruginosa* PAO1 were cultured in LB medium and preserved at − 80 °C for further use.

### Isolate typing and phenotypic identification
Genomic DNA of overnight cultured *P. aeruginosa* isolates were harvested using Bacterial DNA Isolation Kit (Foregene Biotechnology, Co. Ltd., China). Typing of *P. aeruginosa* isolates was performed by ERIC-PCR using the single primer 5′-AACTAAGTAACTGGGGTGAGCG-3′[43]. Phenotypic identification of *P. aeruginosa* isolates was performed as recommended by Filloux and Ramos[44] using PAO1 as positive control. In brief, *P. aeruginosa* was inoculated on M9-skim milk (0.5%, w/v) plates to test the production of QS-controlled extracellular proteases by checking the size of proteolytic halo. Pyocyanin production was determined by extracting pyocyanin from the supernatant using chloroform and HCl, followed by measuring the absorbance at 520 nm. Biofilm production was determined by crystal violet staining followed by measuring the absorbance at 590 nm. The production of siderophores was determined by measuring the cell densities of *P. aeruginosa* cultured in M9-CAA medium supplemented with 100 μg/mL of Transferrin for 24 h. All the experiments were independently repeated for three times. The phenotypic values of *P. aeruginosa* clinical isolates were compared to those of PAO1.

### WGS, genomic and phylogenetic analyses
Libraries of *P. aeruginosa* genomic DNAs were constructed by using NEBNext®Ultra™ DNA Library Prep Kit for Illumina (New England Biolabs, USA), and then WGS was performed on the Illumina HiSeq PE150 platform (Novogene Bioinformatics Technology Co. Ltd., China). The raw data are deposited in the NCBI BioProject database under

accession number PRJNA846307. High quality pair-end reads were assembled by using the software SOAP denovo v2.04, SPAdes, and ABySS, and integrated by CISA, followed by sequence optimization using Gapclose v1.12[45–48]. The program GeneMarkS was used to retrieve the coding sequences, and the gene functions were predicted by using the databases of GO, KEGG, COG, NR, TCDB and Swiss-Prot[49]. MUMmer and LASTZ were used to identify the SNPs and InDels in the genomes of *P. aeruginosa* clinical isolates from the later sampling periods compared to those of the corresponding isolates from the initial[50]. SnpEff v4.3 was used to evaluate the mutation impact of SNPs[51]. The common and different numbers of SNPs, InDels and genes were sorted by using Venn diagram (http://bioinformatics.psb.ugent.be/webtools/Venn/). The assembled contigs of *P. aeruginosa* sequenced in this study and the complete genome sequences of *P. aeruginosa* PAO1 (NCBI accession no. AE004091.2), PA14 (NCBI accession no. NC_008463.1), PAK (NCBI accession no. CP020659.1), DK2 (NCBI accession no. NC_018080.1), and PA7 (NCBI accession no. NC_009656.1) were used to construct SNP-based phylogenetic tree with kSNP v3.0[52]. The genome sequence of *Azotobacter vinelandii* (NCBI accession no. NC_012560.1) was set as the outgroup. The tree was visualized with FigTree v1.4.3 (http://tree.bio.ed.ac.uk/software/figtree/). Genome sequence-based typing of MLST and serotype were performed by using the online software MLST v2.0 (https://cge.food.dtu.dk/services/MLST/) and PAst v1.0 (https://cge.food.dtu.dk/services/PAst/).

### Transcriptomic analysis of *P. aeruginosa* isolates

The total RNAs of *P. aeruginosa* isolates (cultured to cell densities of $OD_{600} = 1.5$ in LB broth) from different sampling periods of COPD patients were isolated using Total RNA Isolation Kit with gDNA removal (Foregene Biotechnology, Co. Ltd., China). Each RNA sample from three independent biological replicates were mixed. RNA samples from two parallel experiments were conducted for library construction using NEBNext®Ultra™ RNA Library Prep Kit for Illumina, followed by prokaryotic strand-specific RNA-seq on the Illumina HiSeq PE150 platform (Novogene Bioinformatics Technology Co. Ltd., China). The raw data are deposited in the NCBI BioProject database under accession number PRJNA846307. High-quality pair-end reads were mapped to the genome of *P. aeruginosa* PAO1 by Bowtie 2 v2.2.3[53]. Differential gene expression was calculated by HTSeq v0.9.1 and DESeq 2 using expected number of fragments per kilobase of transcript per million fragments (FPKM)[54,55]. Combined application of KOBAS v2.0, GOseq R package, and DAVID v6.8 were used to get the functional categories of KEGG pathway and GO enriched by the differentially expressed genes[56–58]. The common and different numbers of differentially expressed genes, the enriched KEGG and GO terms among isolates were sorted by using Venn diagram.

### Competition assay

The interactions of *P. aeruginosa* clinical isolates W1a-ST357, W4b-ST357-*Exp*⁻, W8a-ST357-*Exp*⁻*Pvd*⁻, and W1c-ST1129-*Exp*⁻*Pvd*⁻ from patient W were studied by coculturing different combinations of them under different conditions. The growth status of monocultured isolates in QS-required and not required media was determined also. For LasR-controlled elastase-related two-player's competition assay, W1a-ST357 and W1c-ST1129-*Exp*⁻*Pvd*⁻, W1a-ST357 and W4b-ST357-*Exp*⁻, W1a-ST357 and W8a-ST357-*Exp*⁻*Pvd*⁻, or W1c-ST1129-*Exp*⁻*Pvd*⁻ and W4b-ST357-*Exp*⁻ were cocultured in 2 mL of M9-casein medium from different initial ratios (99:1, 1:1, and 99:1) for 24 h with shaking. For PvdS-controlled siderophore-related two-player's competition assay, W4b-ST357-*Exp*⁻ and W8a-ST357-*Exp*⁻*Pvd*⁻, W1a-ST357 and W1c-ST1129-*Exp*⁻*Pvd*⁻, W1a-ST357 and W8a-ST357-*Exp*⁻*Pvd*⁻, or W1c-ST1129-*Exp*⁻*Pvd*⁻ and W8a-ST357-*Exp*⁻*Pvd*⁻ were cocultured in 2 mL of M9-CAA medium supplemented with Transferrin (100 µg/mL) from an initial ratio of 99:1 for 24 h with shaking. For the three-player's competition assay, a mixture of W1a-ST357, W8a-ST357-*Exp*⁻*Pvd*⁻ and W1c-ST1129-

*Exp*⁻*Pvd*⁻ at the ratio of 98:1:1 was inoculated in 2 mL of M9-casein medium or the medium supplemented with 50 µM of FeCl₃ or 100 µg/mL of Transferrin, and then successively subcultured (1:10 dilution) in accordant fresh medium at 24 h interval. The experiment was stopped when the frequency of each isolate in the culture was relatively stable. All the experiments above were also repeated by using the isolates N1a, N5c-*lasR*, and N7d-*lasRpvdS* from the patient N, or by culturing the four isolates from patient W in ASM[59]. All the experiments were independently repeated for six times. The total cell density of each culture was measured by CFU enumeration on LB plates, and the proportion of each isolate in the coculture was determined by spreading the same volume of culture liquids on different selection plates. Specifically, W1a-ST357 and N1a produced a large and apparent proteolytic ring on M9-casein plate, while W1c-ST1129-*Exp*⁻*Pvd*⁻ and N7d-*lasRpvdS* produced no proteolytic ring. Only W8a-ST357-*Exp*⁻*Pvd*⁻ showed an Imipenem resistance and could be selected on LB-Imipenem (5 µg/mL) plate. The relative fitness ($v$) of one kind of isolates was calculated using the equation $v = \log_{10}[x_1(1 - x_0)/ x_0(1 - x_1)]$, where $x_0$ indicates the initial frequency and $x_1$ indicates the final, and $v > 0$ indicates one isolate grows faster than the opponent, while $v < 0$ indicates an opposite growth status, as described elsewhere[9].

### *Caenorhabditis elegans* killing assay

The pathogenicity of *P. aeruginosa* isolates W1a-ST357, W1c-ST1129-*Exp*⁻*Pvd*⁻, and mixture (1:1) of them were determined by using *C. elegans* slow-killing assay as described previously[21]. In brief, each *P. aeruginosa* solution was spread on nematode growth medium and incubated at 37 °C for 24 h. Subsequently, 15 newly cultured nematodes at L4 stage were seeded on each plate and further incubated at 25 °C. The survival of nematodes was recorded at 24 h interval. Nematodes fed with *Escherichia coli* OP50 (uracil auxotrophy) were set as control.

### Mouse models

Agar-beads-embedded *P. aeruginosa* isolates W1a-ST357, W1c-ST1129-*Exp*⁻*Pvd*⁻, and mixture (1:1) of them, as well as a mixture of W1a-ST357, W8a-ST357-*Exp*⁻*Pvd*⁻, and W1c-ST1129-*Exp*⁻*Pvd*⁻ (98:1:1) were prepared and used to chronically infect the lung of C57BL/6 mouse (8-week-old, female) as previously described[14]. The agar beads were dispersed into 0.5–2.0 × 10⁶ CFUs in 50 µL of sterile saline and intranasally instilled into anaesthetized mice. The survival of mice was recorded at 12 h interval. Three randomly selected mice were killed at designated sampling points. The whole lungs were aseptically removed, homogenized, and conducted for CFU enumeration, RNA isolation, and flow cytometry.

### Flow cytometry

The lung tissues were homogenized in RPMI-1640 and 10% FBS containing 0.2 mg/ml collagenase type I/IV and single-cell suspensions were generated from lung tissues using a gentle MACS™ Octo Dissociator with Heaters following the manufacturer's protocol (Miltenyi Biotec). The digested lung tissues were filtered through 70 µM cell strainers and red blood cells were lysed using red blood cell lysis buffer (Beyotime). The resulting single-cell suspensions were washed with PBS and resuspended in PBS containing 1% FBS. The cell suspensions were incubated with FcR Blocking Reagent (Biolegend) for 15 min at 4 °C, stained with fluorescent conjugated Abs for 30 min at 4 °C and washed twice with PBS. Positive staining with the BD fixable viability dye FVS620 (BD Biosciences) was used to exclude dead cells. Data were collected on a BD FACSymphony analyzer (BD Biosciences) and analyzed with FlowJo software (v.10.4).

### Quantitative protein mass spectrometry

The proteins in the BALs of patients L and W, and in the sputum samples of patient C were collected and conducted for label-free LC-MS/MS using Orbitrap Exploris 480 matched with FAIMS (Thermo Fisher) with

ion source of Nanospray FlexTM (ESI). All the resulting spectra were separately searched against UniProt database for *Homo sapiens* and *P. aeruginosa* by Proteome Discoverer 2.4 (Thermo). The identified peptide spectrum matches and proteins with FDR no more than 1.0% were retained. The protein quantitation results were statistically analyzed by *t*-test. GO and InterPro functional analysis were conducted using the Interproscan program against the non-redundant protein database (including Pfam, PRINTS, ProDom, SMART, ProSite, PANTHER), and the databases of COG (Clusters of Orthologous Groups) and KEGG were used to analyze the protein family and pathway. The common and different numbers of differentially expressed proteins, the enriched KEGG and GO terms among isolates were sorted by using Venn diagram.

### Transcriptomic analysis of host lungs

The total RNAs of mouse lungs and BALs of COPD patient N were isolated and conducted for eukaryotic RNA-seq on the Illumina HiSeq PE150 platform. The raw data are deposited in the NCBI BioProject database under accession number PRJNA846307. High quality pair-end reads were mapped to the genome of *Mus musculus* or *Homo sapiens* by Hisat2 v2.0.5[60]. Differential gene expression was calculated by feature-Counts v1.5.0-p3 and DESeq 2 v2.2.3 using expected number of FPKM[55,61]. Combined application of clusterProfiler R package and DAVID v6.8 were used to get the functional categories of KEGG pathway and GO enriched by the differentially expressed genes[58,61]. The common and different numbers of differentially expressed genes, the enriched KEGG and GO terms among isolates were sorted by using Venn diagram.

### Statistical analysis

Data were processed and visualized by Graphpad Prism v9.0 (San Diego, CA, USA). Mean values of standard deviation were compared by using two-tailed unpaired *t*-test or one-way ANOVA with Tukey post-hoc tests using a 95% confidence interval. The survival curves of *C. elegans* and mice were compared by using Log-rank (Mantel-Cox) test.

### Reporting summary

Further information on research design is available in the Nature Portfolio Reporting Summary linked to this article.

## Data availability

The high-throughput sequencing data are deposited in the NCBI BioProject database under accession number PRJNA846307. Additional data that support the findings of this study are provided in the supplementary files. Source data are provided with this paper.

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

## Acknowledgements

This work was supported by the National Natural Science Foundation of China (32270121 and 31970131 to K.Z., 81922042 and 82172285 to X.Z.), the Sichuan Science and Technology Program (2021JDJQ0042 to K.Z), the high-level talent training program of Chengdu University (2081920066 to K.Z), the 1·3·5 project of excellent development of discipline of West China Hospital of Sichuan University (ZYYC21001 to X.Z.), the innovation foundation of the Affiliated Hospital of Chengdu University (CDFYCX202209 to X.Z.), and the National Key Research and Development Program of China (No. 2023YFC2300033 to X.Z.).

## Author contributions

K.Z. and X.Z. designed, initiated, and conceived the project. K.Z., X.Y., Yi.Z., He.L., J.S.L., Hu.L., Y.W., and T.H. performed the experiments. K.Z., Q.Z., G.H., Ya.Z., and H.Z. coordinated clinical sample collection. K.Z., C.Y., L.D., and X.Z. performed computational analyses. K.Z., X.Y., He.L., J.S.L., and X.Z. assisted with mouse experiments and prepared mouse samples. K.Z., X.Y., and Y.C. assisted with the nematode infection experiments. X.W. and Y.C. provided critical reagents and equipment. K.Z. and X.Z. wrote the manuscript with extensive input from all authors. All authors have read and approved the final manuscript.

## Competing interests

The authors declare no competing interests.
