## [Peer Review File · Nature Communications]

Evolution of *lasR* mutants in polymorphic
Pseudomonas aeruginosa populations facilitates chronic
infection of the lungReviewer #1 (Remarks to the Author):

This is an interesting manuscript by Zhao and colleagues tracking the emergence and behaviors of lasR mutants in people who have chronic *P. aeruginosa* infections. The most interesting observation is the idea that the presence of both WT and LasR mutant cells attenuates the pathogenicity of the bacteria in a way that cannot be explained by either type in a mono-infection (Fig 4).

Major issues:

- The authors have chosen to interpret their data through the prism of cooperation and cheating in bacterial populations, focusing on public goods. They provide little evidence, if any, that is consistent with the idea that there is a social dilemma in the lungs of people with COPD. In fact, there is some evidence (eg, PMID: 24866798) that in the context of lung infections, lasR mutants are better adapted to the environment and are not cheaters.
- Consistent with this issue it is not clear how the experiments in Figure 3 are relevant to the evolution within the lung. That the mutations occur in a sequential way that stabilizes cooperation in laboratory conditions does not mean that this was the selective pressure for the mutations.
- The SNP data and the trees drawn suggest the possibility that some of the patients might have harbored multiple lineages of *P. aeruginosa*, which makes the analysis of mutational change over time rather difficult.

Minor points:

- The figures are hard to read and I had difficulty distinguishing between some of the colors (and figuring out what the control group was)
- Did the authors adjust their statistical tests for multiple comparisons in the cases where there are multiple comparisons?
- The introduction should be reorganized to introduce quorum sensing and QS regulation of extracellular products before a discussion of social cheating.

Reviewer #2 (Remarks to the Author):

The authors collect longitudinal samples of *P. aeruginosa* from COPD patients and look at the evolution of social behaviors, with a primary focus on QS and pyoverdine production. They perform in vitro competition experiments and look at host colonization and immune response in a mouse model, and gene expression in human samples also subjected to mass spec. They infer that the sequential evolution of social cheats on different traits can stabilize a heterogeneous population and affect host response.

It is great to see this type of analyses on clinical isolates. However, I found the manuscript difficult to read, with many evolutionary statements that I don't understand, or that are wrong. Examples are (see also below):

"modifying the adaptability" eg. l 85: we hypothesize that further evolution of the polymorphic *P. aeruginosa* population may mainly concentrate on modifying the adaptability of lasR mutants to facilitate the persistent colonization of the whole population -

and l 233: while the adaptability of lasR mutants would be adjusted by the accumulation of other mutations during further evolution in COPD airways - what does this mean?

130-134: These results suggested that QS-controlled extracellular proteases might be a vitally important public goods that could bring more direct benefits (in terms of nutrient acquisition) for

the survival of *P. aeruginosa* in vivo, followed by siderophores, and the intraspecific competitions of which would be more readily to cause the sequential invasion of individuals deficient in producing the products. – unclear to me

L.465-466: will be introduced to stabilize the population structure and to promote the population fitness during chronic lung infection. Selection doesn't work to stabilize population structure and population fitness.

The naming of isolates makes it difficult to remember what is what, and to see that not all isolates from the same patient are of the same clone type – and therefore not sequentially evolved from the same ancestor. Consider renaming strains to also include clone type, and perhaps even phenotype?

I truly appreciate the challenges of working with clinical isolates. However, the generality of the findings is not clear: the sequential accumulation of mutants in first *lasR* and then *pyoverdine* is based on two patients, where selected isolates of one are analysed in more detail. Additionally, the isolates analysed in co-culture were not sampled at the same time and only assumed to be co-occurring. This is not entirely unreasonable as they were collected within a relatively short time frame, however, it needs to be stated explicitly.

Suggestions for edits:

25: heterogeneous population, not individuals

25-27: unclear, consider rephrasing

34: is cascaded public goods game and established expression?

43: only some can invade hosts

43: delete a kind of

49: invasion instead of sabotage

52: bacterial populations are not colonized

52: detected instead of occurs?

61: experimental instead of laboratorial?

74: under experimental conditions?

78: depends on populations structure!

128-130: add sample sizes

141-142: clarify

181: what is dominant phylogenetic status

207-211: missing stats on which gene categories are hit

225: how are virulence related genes defined

238: mention that w1c is a different clone type – esp when identifying mutations by comparing to a

different clone type

1.245 and 255: stats?

258: where is population crash shown?

272: environmental conditions?

275: unclear : occur during further evolution

301: infection models represent acute not chronic infection

310: unclear: less fluctuated residual cell numbers

313: strains not individuals

317: quantification of damage?

322: differences instead of changes?

324: fluctuations?

352: round 3?

360: difference?

374: express / harbor in stead of evolve

375: rephrase frequently select the mutants

377: delete among

378: can be mutants without being cheaters

387: rephrase are beneficial to the survival advantage

393: unclear rather than further evolutionary selection on lasR mutant that harboring additional resistance-related mutation/s.

405: how where they similar?

407: delete as well

408: harmful to whom?

420: phase transition?

424: unclear and the frequencies of which differed along with the change of environmental cues

427: strong statement for *P. aeruginosa*!

429: mainstream social interaction ?

430: adjusting the roles of lasR mutants ?

431: according to environmental change, and thus provides an explanation for the persistent colonization and QS-related recurrent attacks of *P. aeruginosa*. ??

442: collectively imply the functional diversity of lasR mutants in the polymorphic *P. aeruginosa* population. ?

453: To achieve the above purpose, this may also be a direction of lasR mutant evolution ?

455: and the evolution of multiple independent routes of lasR mutants in COPD can promote the colonization of multiple strains in different infection niches.

487: unsuccessful instead of unnecessary?

490: what is meant with: The single colony of identified

505: DNA singular

518: was siderophore production measured in iron limited or iron supplemented media?

560: mention also monocultures

Figure 2B: what are the black arrow heads? Where do the boxes belong to?

Figure 2B & Table S3: Why compare all isolates back to w1a when some are different clone types? In the case of transmission you could compare these with each other? 801: when comparing two different clone types I don't think it makes sense to list the SNPs etc as accumulated mutations

Figure 4: in the W1C and W1A mono-infections, a number of animals die between day 2-4 – could this impact the CFU counts in B?

Figure 4B & C: I would argue that W1c alone does best – all hosts are alive and CFU counts are steady – so is it really a cheat in vivo?

Figure S1: what are the sample sizes at the different time points?

In patient W, all isolates still produce pyoverdine? How do they grow in iron limited media?

Figure S5: 8 replicates in C?

Fig S7: iron limitation

Reviewer #3 (Remarks to the Author):

Zhao et al. have investigated social cheating of *P. aeruginosa* during chronic infection in the lungs of patients with COPD, specifically focusing on the role of lasR mutants. This study uses a great number of techniques to explore their research question, and the use of longitudinal samples from the same patients is really interesting. However, it would be strengthened by more in depth analysis of some of the data generated to provide more novel insight into COPD *P. aeruginosa* infections over time.

Major comments:

- It would be good to include a table with information for the mutations identified with WGS – which genes are these in? Are all the numbers reported for non-synonymous mutations? Just because there are different numbers of SNPs/indels, it doesn't necessarily mean that these are more important without knowing the location and effect of these mutations.
- The quality of the genome assemblies for the isolates used as a reference to call SNPs for the later isolates needs to be reported. If these are a poor quality or the quality greatly differs between patient reference isolates, this could be a source of variation for the data.
- L155 – what is meant by mutations harmful to gene function? There is little information of the specific mutations found outside of lasR, I think more evidence is needed to make this statement.

- The phylogeny and groups are confusing – *P. aeruginosa* phylogenetic groups are well reported, with PAO1 being part of group 1 and PA14 part of group 2, the groups assigned in this study are not consistent with this. It would also be useful to include the PA7 reference strain in the phylogeny, as it is part of the 3rd phylogenetic group, to see where the isolates in this study sit in the phylogeny in relation to this strain, and would put the isolates from this study into context with standard *P. aeruginosa* phylogeny groups.
- Why were isolates from only one patient selected to phenotypically investigate? It would strengthen the work to look across multiple patients to confirm this isn't only seen in one case.
- It would be useful to repeat competition assays in a more realistic medium (i.e. sputum mimicking), as *P. aeruginosa* behaves differently in different in vitro conditions, and standard laboratory media is known to be a poor representation of chronic *P. aeruginosa* infection.

Minor comments:

- The introduction section has some very long sentences (e.g. L88-92), it would be much easier to read if these were broken up into smaller sentences.
 - If patients were negative for PA for the last year, why were these selected and how was PA isolated from samples of PA-negative patients?
 - Why were isolates selected from those specific patients for further analysis? This needs to be explained, even if it was just at random it would be good to state how this selection was performed.
 - The isolate IDs are quite hard to follow, it would be easier to understand the overall manuscript if they were more intuitive, or include a table with information for each isolate ID to refer back to whilst reading.
 - More method details are needed for the RNA seq analysis from the mouse experiments, the aligners etc. used for bacterial RNA seq analysis are typically not suitable for mammalian RNA seq analysis.
 - Need to add references for the bioinformatics packages/programmes used.
- L83 – better to say 'populations, comprising of' rather than 'population comprises'
- L93 – think this line is missing the word population
- L105 – define BALs acronym here as it is the first time it is used and may not be known by all readers
- Fig 1c – bigger graphs would be easier to read - maybe 2x2 instead?

Authors' point-by-point responses to reviewers' comments:

Reviewer #1 (Remarks to the Author):

This is an interesting manuscript by Zhao and colleagues tracking the emergence and behaviors of *lasR* mutants in people who have chronic *P. aeruginosa* infections. The most interesting observation is the idea that the presence of both WT and LasR mutant cells attenuates the pathogenicity of the bacteria in a way that cannot be explained by either type in a mono-infection (Fig 4).

Authors' responses: We highly appreciate the warm and insightful comments and constructive suggestions of this expert reviewer, and also sincerely thank him/her for the time and effort in reviewing this manuscript.

Major issues:

1. The authors have chosen to interpret their data through the prism of cooperation and cheating in bacterial populations, focusing on public goods. They provide little evidence, if any, that is consistent with the idea that there is a social dilemma in the lungs of people with COPD. In fact, there is some evidence (eg, PMID: 24866798) that in the context of lung infections, *lasR* mutants are better adapted to the environment and are not cheaters. Consistent with this issue it is not clear how the experiments in Figure 3 are relevant to the evolution within the lung. That the mutations occur in a sequential way that stabilizes cooperation in laboratory conditions does not mean that this was the selective pressure for the mutations.

Authors' responses: Thanks for the comments. Yes, compared to the relatively stable and controllable laboratory conditions, clearly proving the detail of bacterial public goods game *in vivo* is still a challenge. By developing an *ex vivo* porcine lung (EVPL) model, Harrison et al. (PMID: 24866798) reported that the *lasR* mutant grew better than the wild-type (WT) *P. aeruginosa in vivo* but showed no significant growth advantage during co-infection. However, we think that the statement of "*lasR* mutants are better adapted to the lung environment and are not cheaters" still remains to be discussed. First of all, the better growth of isogenic *lasR* mutant with deficient quorum-sensing (QS) system than WT *P. aeruginosa in vivo* is kind of counter-intuitive, and this might be due to the absence of an active immune system in the EVPL model. Furthermore, the application of 1:1 (or 60% WT/40% *lasR* mutant) co-infection in EVPL model can better verify the coexistence of *lasR* mutant and WT strains, but may be inappropriate to study the invasion of WT *P. aeruginosa* by *lasR* mutant. A previous study by Rumbaugh et al. (2009. *Current Biology* 19, 341–345) had clearly demonstrated the *in vivo* invasion of *lasR* mutant towards WT *P. aeruginosa*. They found that the frequency of *lasR* mutant during the co-infection with WT *P. aeruginosa* would increase from 1% to 10%, or even to 35% in the chronic infection. Moreover, the work of Köhler (2009. *Proc Natl Acad Sci U S A.* 106:6339-6344) and Andersen (2015. *Proc Natl Acad Sci U S A.* 112:10756-10761) also provide clinical evidence to show the cheating of *lasR* or *pvdS* mutants in the presence of wild-type cells.

The experiments in Fig. 3 of our present study explored the *in vitro* competition of the isolates W1a, W4b, and W8a, which belong to the same lineage from the same patient and carry the mutations occurred in a sequential way. The isolate W1c from the same patient was included as a highly evolved *lasR* mutant that carried several additional mutations, albeit it belongs to a different lineage. These results demonstrated the influence of the sequentially occurred mutations on the interaction of *P. aeruginosa* isolates in the framework of public goods game. We admit that these experiments just provide a paradigm of bacterial interaction *in vitro* by using differentially evolved isolates from the host. It is hard (or currently

impossible) to directly identify the transition of bacterial interaction according to the change of selection pressure in the lung environment. Alternatively, we further investigated the dynamic change of the proportion of each isolate during co-culture and co-infection to verify the outcome of the bacterial competition. We found that the subsequently emerged mutants could invade the original isolate, and social cooperation succeeded both *in vitro* and *in vivo*. In the revised version, this conclusion was further validated by repeating all the experiments using the isolates N1a, N5c-*lasR*, and N7d-*lasRpvdS* from the patient N, and by culturing the four isolates from patient W in artificial sputum medium (ASM) (Figs. S10-S14).

2. The SNP data and the trees drawn suggest the possibility that some of the patients might have harbored multiple lineages of *P. aeruginosa*, which makes the analysis of mutational change over time rather difficult.

Authors' responses: Thanks for the comments. Yes, as we have stated in the main text, 'our results clearly demonstrated the cross-transmission of *P. aeruginosa* among patients and confirmed that COPD airways were frequently cocolonized by polymorphic *P. aeruginosa*' (Lines 145-147). After comparative genome analysis, we have clarified the mutational change of each *P. aeruginosa* lineage from patient W and listed the sequentially evolved mutations in Fig. 2b and Table 1.

Minor points:

1. The figures are hard to read and I had difficulty distinguishing between some of the colors (and figuring out what the control group was)

Authors' responses: Thanks for the comments. We have clarified the meaning of the colors and the control group in the figure legends in the revised version.

2. Did the authors adjust their statistical tests for multiple comparisons in the cases where there are multiple comparisons?

Authors' responses: Thanks for the comments. Multiple comparisons were performed by comparing the data to a designated group using one-way ANOVA with Tukey post-hoc tests using a 95% confidence interval.

3. The introduction should be reorganized to introduce quorum sensing and QS regulation of extracellular products before a discussion of social cheating.

Authors' responses: Thanks for the suggestion. The Introduction section has been carefully reorganized as suggested by this expert reviewer.

Reviewer #2 (Remarks to the Author):

The authors collect longitudinal samples of *P. aeruginosa* from COPD patients and look at the evolution of social behaviors, with a primary focus on QS and pyoverdine production. They perform *in vitro* competition experiments and look at host colonization and immune response in a mouse model, and gene expression in human samples also subjected to mass spec. They infer that the sequential evolution of social cheats on different traits can stabilize a heterogeneous population and affect host response.

It is great to see this type of analyses on clinical isolates. However, I found the manuscript difficult to read, with many evolutionary statements that I don't understand, or that are wrong. Examples are (see also below):

Authors' responses: We highly appreciate the warm and insightful comments and constructive suggestions of this expert reviewer, and also sincerely thank him/her for the time and effort in reviewing this manuscript and the positive recommendation. The whole manuscript has been carefully rewritten according to the suggestions of this reviewer.

1. "modifying the adaptability" eg. l 85: we hypothesize that further evolution of the polymorphic *P. aeruginosa* population may mainly concentrate on modifying the adaptability of *lasR* mutants to facilitate the persistent colonization of the whole population - and l 233: while the adaptability of *lasR* mutants would be adjusted by the accumulation of other mutations during further evolution in COPD airways – what does this mean?

Authors' responses: Thanks for the valuable comments. Sorry for bring any inconveniences to this expert reviewer caused by our inappropriate wording. We have revised the sentences as 'we hypothesize that further evolution of the polymorphic *P. aeruginosa* population may mainly concentrate on selecting the *lasR* mutants that can facilitate the persistent colonization of the population' (Lines 75-77), 'The evolution of the polymorphic *P. aeruginosa* was characterized by selecting different *lasR* mutants...(Line 83)', '...the accumulation of other mutations might be associated with the enhanced adaptability of *lasR* mutants during further evolution' (Lines 212), '...is associated with the accumulation of mutations in the *lasR* mutants...(Lines 447), etc.

2. 130-134: These results suggested that QS-controlled extracellular proteases might be a vitally important public goods that could bring more direct benefits (in terms of nutrient acquisition) for the survival of *P. aeruginosa in vivo*, followed by siderophores, and the intraspecific competitions of which would be more readily to cause the sequential invasion of individuals deficient in producing the products. – unclear to me

Authors' responses: Thanks for the valuable comments. Sorry for bring any inconveniences to this expert reviewer because of our unclear writing. This sentence has been revised as 'Therefore, these data revealed that the *P. aeruginosa* isolates deficient in producing the costly and sharable extracellular products were selected in a sequential way. *P. aeruginosa lasR* mutants deficient in producing extracellular proteases might be selected first in COPD airways, while the *lasR* mutants carrying additional mutations in the *pvdS* gene might be selected during further evolution' (Lines 118-122).

3. L.465-466: will be introduced to stabilize the population structure and to promote the population fitness during chronic lung infection. Selection doesn't work to stabilize population structure and population fitness.

Authors' responses: Thanks for the valuable comments. This sentence has been revised as 'The population structure can be stabilized by the formation of a cascaded public goods game mediated by QS-controlled extracellular products and siderophores during chronic lung infection' (Lines 448-450).

4. The naming of isolates makes it difficult to remember what is what, and to see that not all isolates from the same patient are of the same clone type – and therefore not sequentially evolved from the same ancestor. Consider renaming strains to also include clone type, and perhaps even phenotype?

Authors' responses: Thanks for the valuable suggestion. Yes, as we have written in the main text, the *P. aeruginosa* isolates from patient W could be divided into at least three main lineages. To enhance the distinguishability of the isolates used in each experiment, we have renamed the isolates throughout the manuscript by adding the information of MLST and the relevant phenotypes in the revised version.

5. I truly appreciate the challenges of working with clinical isolates. However, the generality of the findings is not clear: the sequential accumulation of mutants in first *lasR* and then pyoverdine is based on two patients, where selected isolates of one are analysed in more detail. Additionally, the isolates analysed in co-culture were not sampled at the same time and only assumed to be co-occurring. This is not entirely unreasonable as they were collected within a relatively short time frame, however, it needs to be stated explicitly.

Authors' responses: Thanks for the positive and valuable comments of this expert reviewer, and sorry for bring any misunderstandings to this reviewer because of our unclear naming on the isolates and writing. Actually, all the isolates used for the evolutionary trajectory analysis and co-culture/co-infection assays in the present study were from the same patient W. The sequentially evolved isolates W1a, W4b and W8a belong to the same lineage. The highly evolved *lasR* mutant W1c, which was co-isolated with W1a, belongs to another lineage. We have renamed the isolates by adding the information of clone type to enhance the traceability of them.

Furthermore, we would like to clarify that there was a huge number of *P. aeruginosa* clones after the clinical samples from each sampling period were cultured on LB plates. Identifying the genetic composition of all the isolates from each sampling period by using whole-genome sequencing is a huge task and too expensive. Thus, it is too difficult to trace the presence of a specific isolate in host lungs over time. As we have stated in the Materials and Methods section, only 'The colonies with apparent differences in shape, color, size and surface states were picked out...'. The isolates from each sampling period of different patients and representing the phenotypes of others in each subclade of the isolate typing were selected for genomic identification. Although we did not prove the co-occurrence of the isolates W1a, W4b and W8a, the detection of isolates producing extracellular protease like W1a, the isolates harbored *lasR* mutation and *lasR-pvdS* double mutation after 4 months in patient W (Fig. 1c-f, and Fig. S1c), might suggest a strong probability of the coexistence of them. Moreover, we further performed a set of parallel experiments by using the isolates from patient N and had comparable genetic and phenotypic characters to those from patient W, and obtained similar results to those presented in Fig. 3 (Fig. S10 and S11).

Suggestions for edits:

25: heterogeneous population, not individuals

Authors' responses: Thanks for the suggestion. We have corrected this in the revised version (Line 20).

25-27: unclear, consider rephrasing

Authors' responses: Thanks for the comment. This sentence has been revised as 'the dynamic change of cell-cell interaction during further evolution of the polymorphic *P. aeruginosa* population and its impact on host immune system still remains elusive' (Lines 20-22).

34: is cascaded public goods game and established expression?

Authors' responses: Thanks for the comment. Did this reviewer mean 'is cascaded public goods game AN established expression'? If so, cascaded public goods game is a new concept in this study to describe the potential transformation of bacterial social interaction from extracellular protease-mediated public goods game to siderophore-mediated. This is deduced by the sequentially evolved *lasR* mutant and *pvdS* mutant. We have revised this sentence as '...by forming an interaction termed cascaded public goods game...' (Lines 28-29).

43: only some can invade hosts

Authors' responses: Thanks for the comment. We have deleted the words 'including human hosts' in the revised version (Line 38).

43: delete a kind of

Authors' responses: Thanks for the suggestion. We have corrected this in the revised version (Line 38).

49: invasion instead of sabotage

Authors' responses: Thanks for the suggestion. We have corrected this in the revised version (Line 57).

52: bacterial populations are not colonized; 52: detected instead of occurs?

Authors' responses: Thanks for the comments. We have revised this sentence as 'population collapse is rarely detected in bacterial populations that are not colonized natural environments or host tissues' (Lines 60-61).

61: experimental instead of laboratorial?

Authors' responses: Thanks for the suggestion. We have corrected this in the revised version (Line 65).

74: under experimental conditions?

Authors' responses: Thanks for the suggestion. We have corrected this in the revised version (Line 65).

78: depends on populations structure!

Authors' responses: Thanks for the important comment. We have deleted the statement of '...instead of a predominating frequency of *lasR* mutant cheaters as predicted by the theory of public goods game' in the revised version.

128-130: add sample sizes

Authors' responses: Thanks for the suggestion. The sample size has been added in the revised version (Line 116).

141-142: clarify

Authors' responses: Thanks for the important comment. This sentence has been revised as '...The isolates from each sampling period of different patients and representing the phenotypes of others in each subclade of the isolate typing...' in the revised version (Lines 128-130).

181: what is dominant phylogenetic status

Authors' responses: Thanks for the comment. We have revised this sentence as 'The isolates from patient W were selected to predict the *in vivo* evolutionary trajectories of *P. aeruginosa*, because they showed an abundant genetic diversity and located in the position close to the main branch of each phylogenetic group' (Lines 152-154).

207-211: missing stats on which gene categories are hit

Authors' responses: Thanks for the comment. In the revised version, we have listed the mutation types and genes in *P. aeruginosa* COPD isolates from patient W compared to the initial isolate W1a in the new Table 1.

225: how are virulence related genes defined

Authors' responses: Thanks for the comment. The list of *P. aeruginosa* virulence genes was accordant to that in the Virulence Factor Database (<http://www.mgc.ac.cn/VFs/>). We have clarified this in the revised version (Line 205).

238: mention that w1c is a different clone type – esp when identifying mutations by comparing to a different clone type

Authors' responses: Thanks for the valuable suggestions. This sentence has been revised as '...the ancestral isolate W1a (herein renamed as W1a-ST357), which belonged to the lineage ST357:O11, the co-isolated W1c (herein renamed as W1a-ST1129-*Exp⁻Pvd^d*), which belonged to the lineage ST1129:O6 with several loss-of-function mutations in pathoadaptive genes...' (Lines 216-219). Moreover, we have also specified the clone type of W1c when describing the identified mutations.

l.245 and 255: stats?

Authors' responses: Thanks for the comment. The results of statistical analyses have been performed in the revised version (Figs. 3a-d and S5b).

258: where is population crash shown?

Authors' responses: Thanks for the comment. This sentence has been deleted in the revised version.

272: environmental conditions?

Authors' responses: Thanks for the suggestion. We have corrected this in the revised version (Line 258).

275: unclear: occur during further evolution

Authors' responses: Thanks for the comment. This sentence has been deleted in the revised version.

301: infection models represent acute not chronic infection

Authors' responses: Thanks for the comment. As we had stated in the Methods section, the infection models established in the present study were chronic infection, rather than acute. The pathogenicity of *P. aeruginosa* isolates W1a-ST357, W1c-ST1129-*Exp⁻Pvd^d*, and mixture (1:1) of them were determined by using *C. elegans* slow-killing assay (to mimic the status of chronic infection) and intranasally instilling agar bead-encapsulated bacterial cells into anaesthetized mice.

310: unclear: less fluctuated residual cell numbers

Authors' responses: Thanks for the comment. This sentence has been revised as '...showed less

fluctuation of residual cell numbers...' in the revised version (Line 309).

313: strains not individuals

Authors' responses: Thanks for the suggestion. We have corrected this in the revised version (Line 313).

317: quantification of damage?

Authors' responses: Sorry for the inaccuracy, what we meant was that W1a and W1c infection induced more infiltration of immune cells. The histological staining is more of a qualitative description of histological morphology, and may be not suitable for quantitative evaluation of tissue damage. We have corrected this in the revised version (Line 316).

322: differences instead of changes?

Authors' responses: Thanks for the suggestion. We have corrected this in the revised version (Line 322).

324: fluctuations?

Authors' responses: Thanks for the comment. We have revised this word as 'responses' in the revised version (Line 325).

352: round 3?

Authors' responses: Thanks for the comment. This sentence has been revised as '...the samples from the third sampling period of patients L and W...' in the revised version (Lines 351-352).

360: difference?

Authors' responses: Thanks for the comment. This sentence has been revised as 'Indeed, the expression levels of the majority of immune-related proteins in the BALs from the fifth and eighth sampling periods were comparable to those from the first period' (Lines 359-361). Moreover, the supporting data for the conclusion here should be Fig. 5i, instead of Fig. 5h in the last version.

374: express / harbor instead of evolve

Authors' responses: Thanks for the suggestion. We have corrected this in the revised version (Line 373).

375: rephrase frequently select the mutants

Authors' responses: Thanks for the comment. This sentence has been revised as '...and the mutants deficient in producing the sharable extracellular products are frequently selected during evolution' (Lines 374-375).

377: delete among

Authors' responses: Thanks for the comment. This sentence has been deleted in the revised version.

378: can be mutants without being cheaters

Authors' responses: Thanks for the comment. The word of 'cheaters' here has been deleted (Lines 375).

387: rephrase are beneficial to the survival advantage

Authors' responses: Thanks for the comment. This sentence has been revised as '...concentrate on selecting *lasR* mutants that can better adapt to the host tissues' (Line 382).

393: unclear rather than further evolutionary selection on *lasR* mutant that harboring additional resistance-related mutation/s.

Authors' responses: Thanks for the comment. This sentence has been revised as '...rather than the selection of *lasR* mutant with sequentially evolved resistance-related mutation/s' (Line 389).

405: how where they similar?

Authors' responses: Thanks for the comment. This sentence has been revised as 'The two subgroups of *P. aeruginosa* isolates were similar to the previously reported CF-adapted *lasR* mutants, which were collected from the later stage of CF airways and carried several loss-of-function mutations in pathoadaptive genes' (Lines 395-398).

407: delete as well. 408: harmful to whom?

Authors' responses: Thanks for the comment. This sentence has been revised as 'the isolate W1c-ST1129-*Exp⁻Pvd⁻*, which might be transmitted from an unknown reservoir and was co-isolated with W1a-ST357, harbored a large amount of loss-of-function mutations...' (Lines 398-400).

420: phase transition?

Authors' responses: Thanks for the comment. This sentence has been revised as 'These results combined with the sequentially harbored mutations in *lasR* and *pvdS* genes during the evolution of *P. aeruginosa* in COPD airways (Fig. S1), indicated that the *in vivo* interaction of *P. aeruginosa* in the polymorphic population might be multifactorial and transformable' (Lines 413-415).

424: unclear and the frequencies of which differed along with the change of environmental cues

Authors' responses: Thanks for the comment. This sentence has been revised as '...the frequencies of which would be changed under different environmental conditions' (Lines 416-417).

427: strong statement for *P. aeruginosa* !

Authors' responses: Thanks for the comment. This sentence has been revised as '...are unlikely to influence the functions of QS system and siderophore production' (Line 419).

429: mainstream social interaction? 430: adjusting the roles of *lasR* mutants? 431: according to environmental change, and thus provides an explanation for the persistent colonization and QS-related recurrent attacks of *P. aeruginosa*. ??

Authors' responses: Thanks for the comment. These speculative statements have been deleted in the revised version.

442: collectively imply the functional diversity of *lasR* mutants in the polymorphic *P. aeruginosa* population. ?

Authors' responses: Thanks for the comment. This sentence has been revised as '...collectively indicate the evolutionary selection on *lasR* mutants that can better adapt to the environments' (Lines 431-432).

453: To achieve the above purpose, this may also be a direction of *lasR* mutant evolution ? 455: and the evolution of multiple independent routes of *lasR* mutants in COPD can promote the colonization of multiple strains in different infection niches.

Authors' responses: Thanks for the comment. These speculative statements have been deleted in the revised version.

487: unsuccessful instead of unnecessary?

Authors' responses: Thanks for the suggestion. We think the word 'unnecessary' here is fine. Because not all the COPD patients enrolled in the present study always need the endoscopic surgery, which was only applied to the patients with breathing difficulties.

490: what is meant with: The single colony of identified

Authors' responses: Thanks for the comment. This sentence has been revised as 'The single colony of *P. aeruginosa* clinical isolates...' (Line 474).

505: DNA singular

Authors' responses: Thanks for the suggestion. We have corrected this in the revised version (Line 489).

518: was siderophore production measured in iron limited or iron supplemented media?

Authors' responses: Thanks for the valuable comment and sorry for the writing error. The production of siderophores was determined by measuring the cell densities of *P. aeruginosa* cultured in M9-CAA medium supplemented with 100 µg/mL of Transferrin (strict iron-limitation medium) for 24 h (Line 499).

560: mention also monocultures

Authors' responses: Thanks for the suggestion. We have added the statement of 'The growth status of monocultured isolates in QS-required and not required media was determined also' in the revised version (Lines 544-545).

Figure 2B: what are the black arrow heads? Where do the boxes belong to?

Authors' responses: Thanks for the comments. The black arrow heads were omitted in the revised version. The meaning of genes in the boxes has been explained in the figure legends in Lines 777-779.

Figure 2B & Table S3: Why compare all isolates back to w1a when some are different clone types? In the case of transmission you could compare these with each other? 801: when comparing two different clone types I don't think it makes sense to list the SNPs etc as accumulated mutations

Authors' responses: Thanks for the valuable comments. After comparative genomic analysis, we had also realized that the isolates from the same patients were belonged to different clone types. We agree that it is inappropriate to compare different types of isolates back to the initial isolates. Therefore, to check the accumulation of mutations in the isolates with the same clone type, we have performed an additional analysis by comparing W4a and W6b to W1c (Subgroup 1) and comparing W5b and W7f to W3a (Subgroup 2). The results are provided in the new Table 1.

Figure 4: in the W1c and W1a mono-infections, a number of animals die between day 2-4 – could this impact the CFU counts in B?

Authors' responses: In the experimental operation of Fig. 4a and Fig. 4b, the different number of agar bead-embedded bacterial cells we used. In Fig. 4a, we administered a lethal dose of bacteria (2.0×10^6 CFUs in 50 µL of sterile saline) and part of the mice died within 2-4 days. While in Fig. 4b, sublethal dose (0.5×10^6 CFUs in 50 µL of sterile saline) was used and only few mice died. We have revised the infected number of bacteria in the part of "Mouse models" in the Method (Line 581).

Figure 4B & C: I would argue that W1c alone does best – all hosts are alive and CFU counts are steady – so is it really a cheat *in vivo*?

Authors' responses: Thanks for the comments. Yes, we also agree that the highly evolved *lasR* mutant

W1c alone might do best, because this isolate also harbored a large set of loss-of-function mutations in several pathoadaptive genes. As we had shown in Fig. 3l, W1c could indeed invade W1a in mouse lungs from an initial proportion of 1% to a proportion comparable to W1a. Therefore, we further studied the influence of the coexistent W1a and W1c on the immune responses of mice. W1a had a strong pathogenic ability, while W1c lost most of the lethality but adapted to persistent colonization. Even W1c caused no death of mice, we were attracted by the finding that either the mono-infection of W1a or W1c caused a remarkable expression change (increase or decrease) of immune-related genes in mice compared to the uninfected control, while the expression change of these genes in the mice of co-infection group was relatively lower. Moreover, the clearance of *P. aeruginosa* isolates (pure extracellular protease-negative) from the lung of patient C after 12 months also indicates that the persistent colonization of *lasR* mutants *in vivo* might require the WT strains.

Figure S1: what are the sample sizes at the different time points?

Authors' responses: Thanks for the comments. The numbers of isolates at different sampling time points of each patient have been provided in the legend of Fig. S1.

In patient W, all isolates still produce pyoverdine? How do they grow in iron limited media?

Authors' responses: Thanks for the important comments. We have just found that these data were missed when we were copying the data from Excel to GraphPad Prism. We have checked all the data and revalidated the phenotypes and genotypes of the isolates and regenerated the figures. Sorry for this mistake and we would like to thank this expert reviewer for his/her careful and rigorous scientific attitude.

Figure S5: 8 replicates in C?

Authors' responses: Thanks for the comment. Yes, this experiment was performed on the 96-well plate with 8 replicates for each culture.

Fig S7: iron limitation

Authors' responses: Thanks for the comment. We have corrected this in the revised version.

Reviewer #3 (Remarks to the Author):

Zhao et al. have investigated social cheating of *P. aeruginosa* during chronic infection in the lungs of patients with COPD, specifically focusing on the role of *lasR* mutants. This study uses a great number of techniques to explore their research question, and the use of longitudinal samples from the same patients is really interesting. However, it would be strengthened by more in depth analysis of some of the data generated to provide more novel insight into COPD *P. aeruginosa* infections over time.

Authors' responses: We highly appreciate the warm and insightful comments and constructive suggestions of this expert reviewer, and also sincerely thank him/her for the time and effort in reviewing this manuscript and the positive recommendation. We have performed a substantial amount of additional experiments as suggested by this expert reviewer, and the whole manuscript has been carefully rewritten.

Major comments:

- It would be good to include a table with information for the mutations identified with WGS – which genes are these in? Are all the numbers reported for non-synonymous mutations? Just because there are different numbers of SNPs/indels, it doesn't necessarily mean that these are more important without knowing the location and effect of these mutations.

Authors' responses: Thanks for the valuable comments. In the revised version, the mutation genes and types in *P. aeruginosa* COPD isolates from patient W were provided in the new Table 1.

- The quality of the genome assemblies for the isolates used as a reference to call SNPs for the later isolates needs to be reported. If these are a poor quality or the quality greatly differs between patient reference isolates, this could be a source of variation for the data.

Authors' responses: Thanks for the valuable comments. The sequencing coverage, depth, and other information have been added to the original Table 1 and moved to Supplementary Dataset S1 in the revised version.

- L155 – what is meant by mutations harmful to gene function? There is little information of the specific mutations found outside of *lasR*, I think more evidence is needed to make this statement.

Authors' responses: Thanks for the valuable comments. Sorry for the unclear writing. The 'mutations harmful to gene function' are the SNPs happened in the start codon, stop codon, and those resulted in a premature stop, and the InDels that caused a frame-shifted reading open frame. In the revised version, we have provided these mutation types and genes in Table 1, and the statement of 'mutations harmful to gene function' has been revised as loss-of-function mutations throughout the manuscript.

- The phylogeny and groups are confusing – *P. aeruginosa* phylogenetic groups are well reported, with PAO1 being part of group 1 and PA14 part of group 2, the groups assigned in this study are not consistent with this. It would also be useful to include the PA7 reference strain in the phylogeny, as it is part of the 3rd phylogenetic group, to see where the isolates in this study sit in the phylogeny in relation to this strain, and would put the isolates from this study into context with standard *P. aeruginosa* phylogeny groups.

Authors' responses: Thanks for the constructive comments and suggestions. We have reconstructed the phylogenetic tree in Fig. 2a by adding the PA7 reference strain. The results showed that PA7 was distributed in the root of tree, while the main structure of the tree and the grouping are generally similar to

the original. Differently, PA14 was distributed in the Group 1 and totally separated from the PAO1 group with high supporting rate. We have mentioned this in Lines 144-145.

- Why were isolates from only one patient selected to phenotypically investigate? It would strengthen the work to look across multiple patients to confirm this isn't only seen in one case.

Authors' responses: Thanks for the valuable comments. We have checked the phenotypes and genotypes of *P. aeruginosa* isolates from another patient N, who was colonized by a set of *P. aeruginosa* parallel to those of patient W, and identified the isolates N1a (intact *lasR* and *pvdS* genes), N5c-*lasR* (*lasR* mutant), and N7d-*lasR**pvdS* (*lasR* and *pvdS* double mutant). The experiments in Fig. 3 were repeated by using the three isolates from patient N, and similar conclusion to the data of Fig. 3 was obtained. The results were arranged in Figs. S10 and S11.

- It would be useful to repeat competition assays in a more realistic medium (i.e. sputum mimicking), as *P. aeruginosa* behaves differently in different in vitro conditions, and standard laboratory media is known to be a poor representation of chronic *P. aeruginosa* infection.

Authors' responses: Thanks for the valuable comments. We have prepared the artificial sputum medium (ASM) as described by Kirchner et al. (2012. J. Vis. Exp. 64, e3857) and repeated the competition assays in this medium. The results showed that although mono-cultured W1a-ST357, W4b-ST357-*Exp*⁻, W8a-ST357-*Exp*⁻*Pvd*⁻, and W1c-ST1129-*Exp*⁻*Pvd*⁻ showed similar growth status in ASM, W4b-ST357-*Exp*⁻, W8a-ST357-*Exp*⁻*Pvd*⁻, and W1c-ST1129-*Exp*⁻*Pvd*⁻ could also invade W1a-ST357 under different coculture conditions from a low initial frequency (1%) (Figs. S12-S14).

Minor comments:

- The introduction section has some very long sentences (e.g. L88-92), it would be much easier to read if these were broken up into smaller sentences.

Authors' responses: Thanks for the comments. In the revised version, the long sentences throughout the manuscript have been carefully rewritten.

- If patients were negative for PA for the last year, why were these selected and how was PA isolated from samples of PA-negative patients?

Authors' responses: Thanks for the comments. In this study, we focused on recruiting the patients who come to the hospital and found to be PA-positive after etiological examination, and then only the patients who were negative for PA for the last year according to their medical records were enrolled in this project for longitudinal collection of PA.

- Why were isolates selected from those specific patients for further analysis? This needs to be explained, even if it was just at random it would be good to state how this selection was performed.

Authors' responses: Thanks for the valuable suggestion. We have specified the selection of patients and isolates in the revised version. *P. aeruginosa* isolates of patient L (80 years old) and patient M (74 years old), who were randomly selected from Group A to represent the patients with acute lung infection, patient W (92 years old) and patient N (81 years old), who represented the patients with chronic lung infection in Group B and received a relatively longer sampling periods, and the sole patient C (66 years old) in Group C, were selected for further analyses (Lines 101-105). The isolates from each sampling period of different patients and representing the phenotypes of others in each subclade of the isolate

typing were selected for whole-genome sequencing (WGS)-based comparative genomic analyses (Lines 128-130).

- The isolate IDs are quite hard to follow, it would be easier to understand the overall manuscript if they were more intuitive, or include a table with information for each isolate ID to refer back to whilst reading.

Authors' responses: Thanks for the valuable suggestion. To enhance the distinguishability and readability of the isolates used in each experiment, we have renamed the isolates throughout the manuscript by adding the information of clone type (MLST) and the relevant phenotypes in the revised version, also as suggested by the other reviewer.

- More method details are needed for the RNA seq analysis from the mouse experiments, the aligners etc. used for bacterial RNA seq analysis are typically not suitable for mammalian RNA seq analysis.

Authors' responses: Thanks for the important suggestion and sorry for the carelessness. We have carefully checked the methods and rewritten the relevant part (Lines 608-616).

- Need to add references for the bioinformatics packages/programmes used.

Authors' responses: Thanks for the suggestion. All the references have been added in the revised version.

L83 – better to say 'populations, comprising of' rather than 'population comprises'

Authors' responses: Thanks for the suggestion. We have corrected this in the revised version (Line 73).

L93 – think this line is missing the word population

Authors' responses: Thanks for the suggestion. We have corrected this in the revised version (Line 83).

L105 – define BALs acronym here as it is the first time it is used and may not be known by all readers

Authors' responses: Thanks for the suggestion. Actually, BALs acronym has been defined in Line 79.

Fig 1c – bigger graphs would be easier to read - maybe 2x2 instead?

Authors' responses: Thanks for the suggestion. These panels have been enlarged in the revised version.

Reviewer #1 (Remarks to the Author):

The authors have addressed most of my concerns with the original manuscript. I appreciate the focus on a lineage where the evolutionary trajectory is clear, which was a major weakness of the prior version. There are a few things to consider still:

1. The title doesn't make sense. How is social cheating modified?
2. I still believe that the authors should be more circumspect about the conclusion that the selection pressure is one of social cheating. Certainly the order of accumulation of mutations seen in this manuscript version is more consistent with that hypothesis, but it is not the only possibility. Again, just because lasR mutants have a fitness advantage in a specific condition does not necessarily make them cheats.
3. Lines 236-289. I appreciate these experiments but it was difficult to keep the various strains, mutants etc clear and understand what the experiments were testing.
4. Line 46. Recommend against calling *P. aeruginosa* a "star species". Maybe instead a "model species"

Reviewer #2 (Remarks to the Author):

Thank you for the revised manuscript. While still a lot to digest the authors have in my opinion significantly improved the readability of the manuscript.

The combination of methodologies and clinical and in vivo work is very impressive. My main concern still stands that the generality of findings are difficult to infer. As I can see it only four isolates exhibit low or no pyoverdine production (fig1D). Two are characterized in a lot more detail and for one of these (w1c) there is no ancestral isolate to deduce the order of mutations. This limitation needs to be spelled out in my opinion.

Throughout there are still phrasings that would benefit from thorough editing, some suggestions below.

Comments and suggestions:

l.123: there are lots of other mutations that cause loss of pyoverdine production. In Fig.1D and Fig. S1 the vast majority of isolates still produce pyoverdine, only four do not or only little as far as I can see. You write that 1/6 of isolates have a pvdS mutation. Do any the pvdS mutants still produce pyoverdine (does not look like this from Fig.1S)? Would it not make more sense to use the phenotypic data for this trait rather than pvdS mutations?

148: Patients can harbor the same clone type without transmission (Marvig et al. 2015 Nature Genetics). Can you confirm that this is actually transmission by a low SNP number?

218: w1c is a different clone type than w1a so a bit confusing to you rename it W1a-ST1129-Exp-Pvd- --better to start w patient ID?

Table 1: why is w1c and w3a compared to w1b still? they are different clonotypes and differences do not represent mutations

"Evolution concentrate on selecting" Consider rephrasing thorough out

Cascaded games: Specify that this is a term that you make up or add reference

I. 21: coexistence of heterogeneous population.

I.38: "living on the planet " unnecessary

45: including in stead of especially

Colonization of *P. aeruginosa* // *P. aeruginosa* colonizing host lungs

48: treatment in stead of clearance

61: rephrase: that are not colonized natural environments or

Reviewer #3 (Remarks to the Author):

Zhao et al. have investigated social cheating of *P. aeruginosa* during chronic infection in the lungs of patients with COPD, specifically focusing on the role of *lasR* mutants. The revisions made by the authors have greatly improved the clarity and strength of their conclusions, however there are a few points that still need to be addressed.

- The patient and isolate IDs are still quite hard to follow, I appreciate that more information about the isolates has been added but this is still hard to keep of track of in terms of the patients outcome. I think this is really useful to be able to follow to track phenotypic data etc. with the patient outcome – maybe something like patient L.A and M.A etc. to show they are from group A who had passed away by ~2 months, or renaming them to A1, A2, B1, B2, C1 to represent their prognosis groups?
- Addition to the supplementary information of a timeline of sample collection for each patient/patient group would make it clearer all the samples you are working with – I was wondering why aren't there more isolates from patient C than W&N in Figure 1 if they lived longer?
- The phylogenetic group numbers are incorrect throughout from L132-148 and in Figure 2a. There are multiple references (e.g. <https://doi.org/10.1093/gbe/evy259>) that detail the phylogenetic groups of *P. aeruginosa* and the common reference strain PA14 is part of phylogenetic group 2. I think the authors are putting the isolates into their own groups, but this needs to be discussed in context of the established phylogenetic groups and then with a different naming structure for their own groupings.
- I think it would be valuable to discuss why only isolates from group B patients (died by ~8 months) have been phenotypically investigated (W & N) – would you expect differences in isolates from patients who had different outcomes?

REVIEWERS' COMMENTS

Reviewer #1 (Remarks to the Author):

The authors have addressed most of my concerns with the original manuscript. I appreciate the focus on a lineage where the evolutionary trajectory is clear, which was a major weakness of the prior version.

Authors' Response: We highly appreciated the reviewer's valuable suggestions and comments, which helped us to improve our original work.

There are a few things to consider still:

1. The title doesn't make sense. How is social cheating modified?

Authors' Response: Thanks for the comment. After careful consideration, the title has been changed as 'Evolution of *lasR* mutants facilitates the stabilization and chronic infection of polymorphic *Pseudomonas aeruginosa* population'.

2. I still believe that the authors should be more circumspect about the conclusion that the selection pressure is one of social cheating. Certainly the order of accumulation of mutations seen in this manuscript version is more consistent with that hypothesis, but it is not the only possibility. Again, just because *lasR* mutants have a fitness advantage in a specific condition does not necessarily make them cheats.

Authors' Response: Thanks for the valuable comment and suggestion. Yes, although we have provided *in vitro* evidence to show the growth advantage of *lasR* mutants is dependent on cooperative behaviors of QS-intact *P. aeruginosa*, clearly proving the existence of this relationship *in vivo* is still a limitation of our present study. We have revised the related sentences (Lines 413-416, 427-429) by also considering the possibility of host adaptation to make the conclusion more concise.

3. Lines 236-289. I appreciate these experiments but it was difficult to keep the various strains, mutants etc clear and understand what the experiments were testing.

Authors' Response: Thanks for the comment. In the last version, we think we have clearly defined the naming of the isolates in Lines 215-223. For example, W1c-ST1129-*Exp⁻Pvd⁻*, the first letter 'W' indicates this isolate was obtained from patient W, '1c' indicates this isolate was one of the isolates from the first round of sampling, 'ST1129' indicates the lineage of the isolates, '*Exp⁻*' indicates this isolate was deficient in producing extracellular proteases, and '*Pvd⁻*' indicates this isolate was deficient in producing siderophores. By doing this, the readers can clearly distinguish the source, lineage and phenotype of the isolates used in the competition assays. The aims of each competition assay had been provided at the beginning of each paragraph (Lines 234-235, 260-263, 276-277) and with the conclusion at the end.

4. Line 46. Recommend against calling *P. aeruginosa* a "star species". Maybe instead a "model species"

Authors' Response: Thanks for the suggestion. We have revised 'star species' with 'model species' in new Line 45.

Reviewer #2 (Remarks to the Author):

Thank you for the revised manuscript. While still a lot to digest the authors have in my opinion significantly improved the readability of the manuscript. The combination of methodologies and clinical and *in vivo* work is very impressive. My main concern still stands that the generality of findings are difficult to infer. As I can see it only four isolates exhibit low or no pyoverdine production (fig1D). Two are characterized in a lot more detail and for one of these (w1c) there is no ancestral isolate to deduce the order of mutations. This limitation needs to be spelled out in my opinion.

Authors' Response: We sincerely appreciated the reviewer's valuable suggestions and comments, which helped us to improve our original work. Actually, we are a little confused by the comment of this expert reviewer about Fig. 1d, because this panel represents the levels of biofilm production, while Fig. 1f represents the percent cell densities of *P. aeruginosa* clinical isolates in iron-limitation medium to PAO1. We agree with this reviewer concerning the generality of the findings. The lifetime of COPD patients who become *P. aeruginosa*-positive is generally short, especially compared with those of CF. Our present study enrolled only two COPD patients (W and N) who was colonized by polymorphic *P. aeruginosa* population and with a sampling period over six months. We believe that further accumulation of *P. aeruginosa* isolates from more these kinds of patients, and we have spelled out this limitation in the Discussion section (Lines 410-412).

Throughout there are still phrasings that would benefit from thorough editing, some suggestions below.

Comments and suggestions:

1. l.123: there are lots of other mutations that cause loss of pyoverdine production. In Fig.1D and Fig. S1 the vast majority of isolates still produce pyoverdine, only four do not or only little as far as I can see. You write that 1/6 of isolates have a *pvdS* mutation. Do any the *pvdS* mutants still produce pyoverdine (does not look like this from Fig.1S)? Would it not make more sense to use the phenotypic data for this trait rather than *pvdS* mutations?

Authors' Response: Thanks for the comments. As we have responded above, Fig. 1d represents the levels of biofilm production, rather than pyoverdine production, and Fig. S1c represents the isolates from patient W. Fig. 1f represents the percent cell densities of *P. aeruginosa* clinical isolates in iron-limitation medium to PAO1.

2. 148: Patients can harbor the same clone type without transmission (Marvig et al. 2015 Nature Genetics). Can you confirm that this is actually transmission by a low SNP number?

Authors' Response: Thanks for the comments. Yes, several evidences have shown that CF patients are frequently colonized by a dominant *P. aeruginosa* lineage. The conclusion of *P. aeruginosa* transmission among COPD patients in the present study was based on the identification of *P. aeruginosa* isolates with different clone types (lineages), phylogenetic status, and distinct number of SNPs compared with the initially obtained isolates from each patient.

3. 218: w1c is a different clone type than w1a so a bit confusing to you rename it W1a-ST1129-Exp⁻Pvd⁻--better to start w patient ID?

Authors' Response: Thanks for the comment. Sorry for the typo. W1c here should be W1c-ST1129-*Exp⁻Pvd⁻*, rather than W1a-ST1129-*Exp⁻Pvd⁻*. We have corrected this typo in the revised version (Line 216).

4. Table 1: why is w1c and w3a compared to w1b still? they are different clonotypes and differences do not represent mutations

Authors' Response: Thanks for the comment. We are confused by the comment 'why is w1c and w3a compared to w1b still' of this reviewer. Actually, W1c and W3a in Table 1 were compared to W1a, because these two isolates had no specific ancestral isolates but had their own 'descendants'. Thus, we first compared W1c and W3a to W1a to show their genetic difference from the W1a lineage, and then compared their 'descendants' (such as W4a, W6b, W5b and W7f) to them to show the evolution of these lineages, respectively.

5. "Evolution concentrate on selecting" Consider rephrasing thorough out

Authors' Response: Thanks for the comment. To avoid ambiguity, the sentences related to 'Evolution concentrate on selecting' throughout the manuscript have been revised (Lines 75-77, 90, 148, 186-187, 295-296, 392-393).

6. Cascaded games: Specify that this is a term that you make up or add reference

Authors' Response: Thanks for the comment. The term of 'cascaded public goods game' has been specified in Lines 289-290.

7. I. 21: coexistence of heterogeneous population.

Authors' Response: Thanks for the comment. This sentence has been revised as suggested.

8. I.38: "living on the planet " unnecessary

Authors' Response: Thanks for the comment. We have deleted 'living on the planet' in the revised version.

9. 45: including in stead of especially

Authors' Response: Thanks for the comment. This sentence has been revised as suggested.

10. Colonization of *P. aeruginosa* // *P. aeruginosa* colonizing host lungs

Authors' Response: Thanks for the comment. This sentence has been revised as suggested.

11. 48: treatment in stead of clearance

Authors' Response: Thanks for the comment. This sentence has been revised as suggested.

12. 61: rephrase: that are not colonized natural environments or

Authors' Response: Thanks for the comment. This sentence has been revised as '...population collapse is rarely detected in bacterial populations that colonize natural environments or host tissues' (Line 60).

Reviewer #3 (Remarks to the Author):

Zhao et al. have investigated social cheating of *P. aeruginosa* during chronic infection in the lungs of patients with COPD, specifically focusing on the role of *lasR* mutants. The revisions made by the authors have greatly improved the clarity and strength of their conclusions, however there are a few points that still need to be addressed.

Authors' Response: We sincerely appreciated the reviewer's valuable suggestions and comments, which helped us to improve our original work.

- The patient and isolate IDs are still quite hard to follow, I appreciate that more information about the isolates has been added but this is still hard to keep of track of in terms of the patients outcome. I think this is really useful to be able to follow to track phenotypic data etc. with the patient outcome – maybe something like patient L.A and M.A etc. to show they are from group A who had passed away by ~2 months, or renaming them to A1, A2, B1, B2, C1 to represent their prognosis groups?

Authors' Response: Thanks for the comment. In the last version, we think we have clearly defined the naming of the isolates in Lines 215-223. For example, W1c-ST1129-*Exp*⁻*Pvd*⁻, the first letter 'W' indicates this isolate was obtained from patient W, '1c' indicates this isolate was one of the isolates from the first round of sampling, 'ST1129' indicates the lineage of the isolates, '*Exp*⁻' indicates this isolate was deficient in producing extracellular proteases, and '*Pvd*⁻' indicates this isolate was deficient in producing siderophores. By doing this, the readers can clearly distinguish the source, lineage and phenotype of the isolates used in the competition assays.

- Addition to the supplementary information of a timeline of sample collection for each patient/patient group would make it clearer all the samples you are working with – I was wondering why aren't there more isolates from patient C than W&N in Figure 1 if they lived longer?

Authors' Response: Thanks for the comment. In fact, we also thought about the addition of the timeline of sample collection at the beginning. However, we found that summarizing the sampling timelines of the remaining 18 enrolled COPD patients ($n = 25$) into one figure was very chaotic and uninformative. Alternatively, the sampling timelines of the five patients mainly analyzed in the present study was merged with the results of bacterial phenotypic identification (the Y-axis of panels c-f in Fig. 1). Moreover, the present study mainly focuses on studying the evolution of *lasR* mutants in the polymorphic *P. aeruginosa* population (Lines 75-77). The result of phenotypic identification revealed that *P. aeruginosa* isolates from patients W and N were the perfect candidates, while there were only QS deficient isolates in patient C and no extracellular protease-positive isolate was identified. Therefore, more isolates were collected from patients W and N.

- The phylogenetic group numbers are incorrect throughout from L132-148 and in Figure 2a. There are multiple references (e.g. <https://doi.org/10.1093/gbe/evy259>) that detail the phylogenetic groups of *P. aeruginosa* and the common reference strain PA14 is part of phylogenetic group 2. I think the authors are putting the isolates into their own groups, but this needs to be discussed in context of the established phylogenetic groups and then with a different naming structure for their own groupings.

Authors' Response: Thanks for the comment. Yes, there are indeed multiple references had detailed the phylogenetic groups of *P. aeruginosa* by using globally collected genomes, including our prior work. However, the majority of the isolates used for constructing the phylogenetic tree were newly collected in

our present study. Only the model strains PAO1, PA14, PAK, DK2, and PA7 were added to indicate the phylogenetic status of these newly collected clinical isolates. Therefore, we just named the groups of these isolates according to the structure of tree, but did not refer to the previous work. We have renamed the structure by using Clade 1, Clade 2, and Clade 3.

- I think it would be valuable to discuss why only isolates from group B patients (died by ~8 months) have been phenotypically investigated (W & N) – would you expect differences in isolates from patients who had different outcomes?

Authors' Response: Thanks for the comment. As we have responded above, the present study mainly focuses on studying the evolution of *lasR* mutants in the polymorphic *P. aeruginosa* population (Lines 75-77). The result of phenotypic identification revealed that *P. aeruginosa* isolates from patients W and N were the perfect candidates (WT and mutant strains co-existed).